# Understanding the Unfairness in Network Quantization

**Bing Liu** [1 2 3]   **Wenjun Miao** [1]   **Boyu Zhang** [1]   **Qiankun Zhang** [✉ 1 2 3]   **Bin Yuan** [1 3 4 5]   **Jing Wang** [6]   **Shenghao Liu** [1]   **Xianjun Deng** [1]

## Abstract

Network quantization, one of the most widely studied model compression methods, effectively quantizes a floating-point model to obtain a fixed-point one with negligible accuracy loss. Although great success was achieved in reducing the model size, it may exacerbate the unfairness in model accuracy across different groups of datasets. This paper considers two widely used algorithms: Post-Training Quantization (PTQ) and Quantization-Aware Training (QAT), with an attempt to understand how they cause this critical issue. Theoretical analysis with empirical verifications reveals two responsible factors, as well as how they influence a metric of fairness in depth. A comparison between PTQ and QAT is then made, explaining an observation that QAT behaves even worse than PTQ in fairness, although it often preserves a higher accuracy at lower bit-widths in quantization. Finally, the paper finds out that several simple data augmentation methods can be adopted to alleviate the disparate impacts of quantization, based on a further observation that class imbalance produces distinct values of the aforementioned factors among different attribute classes. We experiment on either imbalanced (UTK-Face and FER2013) or balanced (CIFAR-10 and MNIST) datasets using ResNet and VGG models for empirical evaluation.

[1]School of Cyber Science and Engineering, Huazhong University of Science and Technology, Wuhan, China [2]Key Laboratory of Cyberspace Security, Ministry of Education, Zhengzhou, China [3]Hubei Key Laboratory of Distributed System Security, Wuhan, China [4]Songshan Laboratory, Zhengzhou, China [5]Visiting researcher with the Lion Rock Labs of Cyberspace Security, CTlHE, Hong Kong, China [6]School of Software Engineering, Huazhong University of Science and Technology, Wuhan, China. Correspondence to: Qiankun Zhang <qiankun@hust.edu.cn>.

*Proceedings of the 42$^{nd}$ International Conference on Machine Learning*, Vancouver, Canada. PMLR 267, 2025. Copyright 2025 by the author(s).

## 1. Introduction

In recent years, with the advancements in computer performance and the maturation of data processing technologies, deep neural networks have made significant strides in fields such as computer vision and natural language processing, achieving impressive results. However, high computational time consumption and large memory overheads pose significant challenges to the efficient implementation of deep neural networks on resource-limited devices. To address these challenges, *neural network quantization* (Huang et al., 2024; Jha et al., 2024; Gholami et al., 2022) is one of the highly effective methods to reduce the power and latency of neural network inference.

To achieve these savings, quantization stores weights and activation tensors as low-bit fixed-point numbers (e.g. 4 or 8-bit) instead of original 32-bit floating-point representation. This greatly reduces data storage requirements, as well as the size and energy consumption of MAC operations, speeding up network execution. There are generally two main classes of algorithms: Post-Training Quantization (PTQ) (He et al., 2024; Yao et al., 2022) and Quantization-Aware Training (QAT) (Xie et al., 2024; Nagel et al., 2022). While PTQ quantizes the model after training and requires no retraining, QAT requires fine-tuning and access to training data. Notably, Nagel et al. (2021) show that both methods do not suffer significantly in terms of model accuracy when compared to their original floating-point counterparts.

Although quantization causes little degradation to overall accuracy on a test set, previous studies (Hooker et al., 2019; 2020) observe that disproportionately high errors may appear among different groups of datasets. As a first glimpse, we experiment on a facial recognition task with UTK-Face dataset (Zhang et al., 2017) and VGG19 model (Simonyan & Zisserman, 2014), focusing on the accuracy of quantized models by PTQ and QAT from 32-bit to 4-bit.

We summarize our key observations in Figure 1 as follows:

- While models are unfair in accuracy before quantization among different groups of individuals, quantization exacerbates such unfairness as bit-widths get lower. For example, 4-bit QAT barely decreases accuracy in group White, but group Others suffers a dramatic drop.

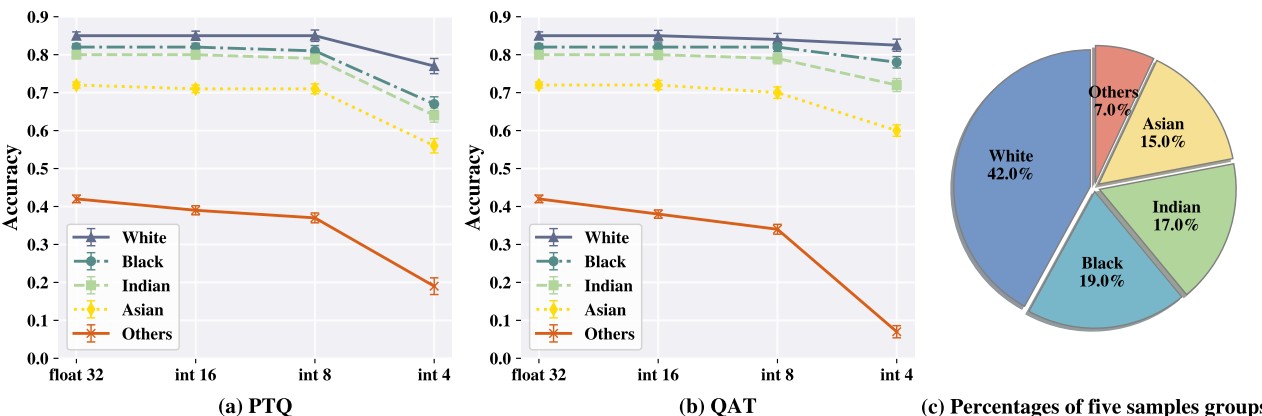

*Figure 1.* Experiments on UTK-Face dataset on the accuracy of each subgroup of individuals using ResNet50 for an ethnicity classification task. Both PTQ and QAT are evaluated as bit-widths get lower.

- Though QAT has a better guarantee of accuracy than PTQ, PTQ behaves better in fairness preservation. For example, from 32-bit to 4-bit, accuracy in group Others decreases with a smaller percentage in PTQ than QAT.

These observations should receive great attention, because such facial recognition tasks have been deployed in some resource-limited devices, e.g., mobile phones or access control systems, and they could potentially harm fairness and privacy. This constitutes our biggest motivation in this paper for a better understanding and elimination of such unfairness caused by quantization for equality protection and privacy preservation. An intuitive understanding seems not difficult: quantization involves loss of parameter information by representing weights and activation tensors as low-precision fixed-point numbers, and thus for underrepresented groups, whose available feature information is already relatively limited, the loss of parameter information can further prevent accurate learning of their features. For an in-depth study, our paper makes a step toward answering the following questions:

*Our Research Questions*

- What are the factors that exacerbate the unfairness in model quantization?
- How do these factors influence the disparate proportions of accuracy degradation among different groups of data?
- How can we mitigate such unfairness to address the effect of these factors?

**Our contributions.** We present a theoretical analysis in disparate impacts of both PTQ and QAT on models' fairness separately. Our main findings are:

- accuracy disparities come from two main factors: (1)

the *gradient norm* of the group loss function and (2) the *trace* of the group loss function's Hessian matrix, regardless of whether PTQ or QAT is used,

- class imbalance induces distinct values of both factors among classes. Both the gradient norm and trace of the Hessian matrix *increase* as the size of a subgroup *decreases*, indicating a positive correlation between the two factors,

- the unfairness caused by QAT is *more severe* than PTQ, because besides being influenced by the above two factors separately, QTA is additionally influenced by their interactions.

Based on the theoretical analysis, we empirically validate our findings on different datasets (UTK-Face, FER2013, CIFAR-10 and MNIST) and models (ResNet and VGG). Finally, to mitigate the unfairness caused by quantization and validate our main findings that a balanced dataset is crucial for fairness preservation, a natural idea is to utilize *data augmentation* techniques for training data. Two simple but effective augmentation methods are empirically evaluated for both PTQ and QAT.

## 2. Related Work

As two main keywords of this paper, both quantization (Jha et al., 2024; Yang et al., 2024b; Gholami et al., 2022; Nagel et al., 2021; Jacob et al., 2018) and fairness (Caton & Haas, 2024; Lalor et al., 2024; Zhang et al., 2024a; Mehrabi et al., 2021) are well studied separately. For quantization, extensive efforts have been devoted to improving quantization performance using either QAT (Liu et al., 2023; Nagel et al., 2022; Esser et al., 2020; Choi et al., 2018) or PTQ (Xiao et al., 2023; Frantar & Alistarh, 2022; Wei et al., 2022; Li et al., 2021). For fairness, studies vary in research fields, for example, federated learning (Badar et al., 2024; Li et al.,

2019), graph neural networks (Yang et al., 2024a; Dong et al., 2022), generative adversarial networks (Zhang et al., 2024b; Xu et al., 2018), and etc.

Our work falls under the broad umbrella of studying the social impacts of ML techniques. Much more related to ours, previous studies have observed through empirical experiments that various neural network compression techniques can cause unfairness in different learning tasks, where the classification accuracy of certain categories may be more affected than others (Jha et al., 2019; Joseph et al., 2020). Most of these observations lie in CV tasks. Hooker et al. (2020) use visualization methods to study the disparate biases introduced by quantization and pruning on different datasets; Tran et al. (2022) elucidate the theoretical factors that exacerbate model unfairness due to pruning in face recognition classification tasks and proposes a mitigation strategy; Hooker et al. (2019) compare the different fairness impacts of pruning and quantization and find that high levels of pruning incur a far higher disparate impact than is observed for the quantization techniques; and Blakeney et al. (2021) propose two simple yet effective metrics, Combined Error Variance (CEV) and Symmetric Distance Error (SDE), to quantitatively evaluate the induced bias prevention quality of pruned models and demonstrate that knowledge distillation can mitigate induced bias in pruned neural networks, even with imbalanced datasets. The fairness impact of compression has also been evaluated in NLP tasks. For example, Du et al. (2021) and Xu et al. (2021) measure the robustness of compressed large language models based on experience, while Ahia et al. (2021) study how compression schemes affect data restriction mechanisms. Xu & Hu (2022) investigate a method to improve fairness by compressing the generated language model.

Mitigating unfairness through data augmentation is a widely used strategy, particularly effective in addressing accuracy disparities arising from data imbalances. For datasets encompassing a protected attribute, Sharma et al. (2020) define an "ideal world dataset" as data where different groups within the protected attribute attain the same label, irrespective of other feature values. By implementing this data augmentation technique, it effectively reduces bias in line with two key fairness metrics: statistical parity difference and average odds difference. Furthermore, to improve the generalizability of fair classifiers, Mroueh et al. (2021) propose "Fair Mixup", a data augmentation strategy for imposing the fairness constraint. In particular, they show that fairness can be achieved by regularizing the models on paths of interpolated samples between the groups. However, distinct from the aforementioned methods, based on our theoretical and empirical findings, we extend data augmentation techniques geometric transformation and random erasing (Zhong et al., 2020) by empirically demonstrating the utility of data augmentation in mitigating bias in quantized models, especially

in the realms of QAT and PTQ.

Building upon the experimental observations mentioned above, this article delves deeper into the specific factors behind the degradation of model fairness caused by neural network quantization and provides an effective mitigation strategy to alleviate this unfairness.

## 3. Preliminaries

This section presents necessary background on model quantization, a formal definition of our research problem and the metric we use for fairness measure.

### 3.1. Quantization and Error Bound

**Signed symmetric uniform quantization.** In this paper, we focus on quantizations only on weights $\boldsymbol{w} \in [\boldsymbol{w}_{min}, \boldsymbol{w}_{max}]$ rather than activations. Since the distribution of the neural network parameters is usually symmetric about 0 (Glorot & Bengio, 2010), we assume $\boldsymbol{w}_{min} = -\boldsymbol{w}_{max}$. Thus, we base on a widely-used signed symmetric uniform quantizer, which is parameterized by a scale $s = \frac{\boldsymbol{w}_{max}}{2^{b-1}-1}(0 < s < 1)$ which specifies the step size of the quantizer, and the bit-width $b$. It maps a real-valued vector $\boldsymbol{w}$ to an integer-valued vector $\boldsymbol{w}_{int}$ by [1]:

$$\boldsymbol{w}_{int} = \text{clamp}\left(\lfloor \frac{\boldsymbol{w}}{s} \rceil; -2^{b-1}, 2^{b-1} - 1\right),$$

$$\text{where clamp}(w; a, c) = \begin{cases} a & w < a\,; \\ w & a \le w \le c\,; \\ c & w > c\,. \end{cases} \tag{1}$$

Note that Eqn. (1) maps floating-point 0 to integer 0. Floating-point weight vector $\boldsymbol{w}$ is stored as integral $\boldsymbol{w}_{int}$. An approximation of $\boldsymbol{w}$, denoted as $\widetilde{\boldsymbol{w}}$, can be de-quantized from $\boldsymbol{w}_{int}$ by:

$$\widetilde{\boldsymbol{w}} = s \cdot \boldsymbol{w}_{int} \approx \boldsymbol{w}\,. \tag{2}$$

Combining Eqn. (1) and Eqn. (2) gives a quantization function $q$ from $\boldsymbol{w}$ to $\widetilde{\boldsymbol{w}}$:

$$\widetilde{\boldsymbol{w}} = q(\boldsymbol{w}; s, b) = s \cdot \text{clamp}\left(\lfloor \frac{\boldsymbol{w}}{s} \rceil; -2^{b-1}, 2^{b-1} - 1\right). \tag{3}$$

**Quantization error.** Eqn. (3) indicates a certain error, defined as $\Delta \boldsymbol{w} = \widetilde{\boldsymbol{w}} - \boldsymbol{w}$, between $\boldsymbol{w}$ and $\widetilde{\boldsymbol{w}}$ after quantization. The error comes from two parts. One comes from the $\lfloor \cdot \rceil$ operator, lying within a range of $\left[-\frac{1}{2}s_i, \frac{1}{2}s_i\right]$ (Nagel et al., 2021) for each component $i$ of $\boldsymbol{w}$. The other comes from the clipping error, but our quantization scheme does not introduce this error, as the quantization factor $s = \frac{\boldsymbol{w}_{max}}{2^{b-1}-1}$

---

[1] $\lfloor \cdot \rceil$ is the round-to-nearest operator.

ensures that the quantized integers lie between $-2^{b-1}$ and $2^{b-1} - 1$. Thus the $\ell_2$ norm [2] of $\Delta w$ can be upper bounded by:

$$\|\Delta \boldsymbol{w}\| \leq \frac{1}{2}\sqrt{n}s_{max}^2 \,, \tag{4}$$

where $n$ is the number of dimensions of $\boldsymbol{w}$ and $s_{max}$ is the largest scale used in the quantization among all parameters. See Appendix A.1 for detailed proof of Eqn.(4). We point out that readers may easily verify that $s$ is inversely proportional to $b$, meaning that quantization to lower bits will induce larger errors.

### 3.2. Problem Definition and Fairness Metric

**Empirical risk minimization (ERM).** We consider a classification task over a dataset $\boldsymbol{D}$ that learns a classifier $f_{\boldsymbol{w}} : \mathcal{X} \to \mathcal{Y}$ parameterized by $\boldsymbol{w}$ using ERM. $\boldsymbol{D}$ consists of $m$ individual data points $(\boldsymbol{x}_i, a_i, y_i)$ for each $i \in [m]$, drawn i.i.d. from an unknown distribution. $\boldsymbol{x}_i \in \mathcal{X}$ represents an input feature vector, $a_i \in \mathcal{A}$ represents a (private or protected) attribute of subgroups, and $y_i \in \mathcal{Y}$ represents the label. We take the facial recognition task on UTK-Face in Figure 1 as an example for explanations. $\boldsymbol{x}_i$ is a face photo of an individual, $a_i$ is the ethnicity (White, Black, Indian, Asian, and Others) of each individual, and $y_i$ is also the label of ethnicity attributes. (In this task, we let $\mathcal{A} = \mathcal{Y}$, but they are not necessarily identical.) $\boldsymbol{w}$ is an $n$-dimensional real-valued vector, and it is trained by:

$$\boldsymbol{w}^* = \arg\min_{\boldsymbol{w}} \mathcal{L}(\boldsymbol{w}; \boldsymbol{D}) = \arg\min_{\boldsymbol{w}} \frac{1}{m}\sum_{i=1}^{m}\ell(f_{\boldsymbol{w}}(\boldsymbol{x}_i), y_i) \,, \tag{5}$$

where $\ell : \mathcal{Y} \times \mathcal{Y} \to \mathbb{R}_+$ is a non-negative loss function.

**Metric of fairness.** To measure the fairness impacts caused by quantization, first recall that $\boldsymbol{w}^*$ can only be approximated as in Eqn. (3), the error leads to a difference between risk functions, which is called an excessive loss. That is, for each group $a \in \mathcal{A}$, we define:

$$\mathcal{G}(a) = \mathcal{L}(\tilde{\boldsymbol{w}}^*; \boldsymbol{D}_a) - \mathcal{L}(\boldsymbol{w}^*; \boldsymbol{D}_a) \,, \tag{6}$$

where $\boldsymbol{D}_a$ denotes the subset of $\boldsymbol{D}$ containing exclusively samples whose group attribute $a_i = a$, $\tilde{\boldsymbol{w}}^*$ denotes the quantized model parameters while $\boldsymbol{w}^* = \arg\min_{\boldsymbol{w}}\mathcal{L}(\boldsymbol{w}; \boldsymbol{D})$. Further, fairness can be measured by the largest gap of excessive losses among all pairs of protected attributes in $\mathcal{A}$:

$$\varphi(\boldsymbol{D}) = \max_{a, a' \in \mathcal{A}} |\mathcal{G}(a) - \mathcal{G}(a')| \,. \tag{7}$$

Eqn. (7) gives our metric of fairness. Our main goal in this paper is to study: (1) What factors are responsible for quantized models with $\varphi(\boldsymbol{D}) > 0$? (2) Why does $\mathcal{G}(a)$

vary among all protected attributes $a \in \mathcal{A}$? (3) Do QAT and PTQ behave differently with respect to $\varphi(\boldsymbol{D})$? (4) Any mitigation strategies can be proposed to minimize $\varphi(\boldsymbol{D})$? The following sections address all these issues.

## 4. Fairness in Post-Training Quantization

In this section, we discuss the degradation of fairness caused by PTQ, which takes a pre-trained 32-bit floating-point network and converts it directly into a fixed-point network without fine-tuning. Recall that in Eqn. (1) the weight vector $\boldsymbol{w}$ is first quantized into $\boldsymbol{w}_{int}$, and then calculations are based on the approximation $\tilde{\boldsymbol{w}}$ defined in Eqn. (2), inducing an error bound of $\frac{1}{2}\sqrt{n}s_{max}^2$ in Eqn. (4). We assume that the loss function $\ell$ is twice differentiable, e.g., MSE loss. To see what factors influence the excessive loss $\mathcal{G}(a)$ of a specific group $a \in \mathcal{A}$, the following upper bound [3] is useful:

**Theorem 4.1.** *Let $\ell$ be a twice differentiable loss function and consider $\boldsymbol{w}^*$ is quantized to low bits using PTQ. The excessive loss for group $a \in \mathcal{A}$ is upper bounded by:*

$$\mathcal{G}(a) \leq \frac{1}{2}\sqrt{n}s_{max}^2 \cdot \|\boldsymbol{g}_{\boldsymbol{w}^*}^{\boldsymbol{D}_a}\| + \frac{1}{8}ns_{max}^4 \cdot \mathrm{Tr}(\boldsymbol{H}_{\boldsymbol{w}^*}^{\boldsymbol{D}_a})$$
$$+ \mathcal{O}\Big(\|\Delta \boldsymbol{w}^*\|^3\Big) \,, \tag{8}$$

*where $\boldsymbol{g}_{\boldsymbol{w}^*}^{\boldsymbol{D}_a} = \nabla_{\boldsymbol{w}}\mathcal{L}(\boldsymbol{w}^*; \boldsymbol{D}_a)$ is the vector of gradient associated with the ERM function $\mathcal{L}$ evaluated at $\boldsymbol{w}^*$ and computed using group data $\boldsymbol{D}_a$, $\mathrm{Tr}(\boldsymbol{H}_{\boldsymbol{w}^*}^{\boldsymbol{D}_a})$ is the trace of the Hessian matrix $\boldsymbol{H}_{\boldsymbol{w}^*}^{\boldsymbol{D}_a} = \nabla_{\boldsymbol{w}}^2\mathcal{L}(\boldsymbol{w}^*; \boldsymbol{D}_a)$ of the ERM function $\mathcal{L}$, at the optimal parameter vector $\boldsymbol{w}^*$, computed using the group data $\boldsymbol{D}_a$, and $\Delta \boldsymbol{w}^*$ is the quantized error of the optimal parameter vector $\boldsymbol{w}^*$.*

The main ingredient of our proof for Theorem 4.1 is a second-order Taylor expansion of the objective function $\mathcal{L}$ at $\boldsymbol{w}^*$ with the assistance of several inequalities and the consistency of $\ell_2$ norms between matrices and vectors.

**Relationship to $\varphi(\boldsymbol{D})$.** In Theorem 4.1, except for a negligible term, $\mathcal{G}(a)$ is related to the sum of: (1) the product of error bounds determined by $s_{max}$ and the gradient norm $\|\boldsymbol{g}_{\boldsymbol{w}^*}^{\boldsymbol{D}_a}\|$ for group $\boldsymbol{D}_a$; (2) the product of a term positively correlated to error bounds and the trace of the Hessian matrix $\mathrm{Tr}(\boldsymbol{H}_{\boldsymbol{w}^*}^{\boldsymbol{D}_a})$ for group $\boldsymbol{D}_a$. Let's consider a data group $\boldsymbol{D}_a$ with protected attribute $a$. If the corresponding gradient norm $\|\boldsymbol{g}_{\boldsymbol{w}^*}^{\boldsymbol{D}_a}\|$ and trace of Hessian matrix $\mathrm{Tr}(\boldsymbol{H}_{\boldsymbol{w}^*}^{\boldsymbol{D}_a})$ are *larger* than other attributes, $\mathcal{G}(a)$ *grows faster* as the bit-width $b$ gets lower (because $s_{max} \propto \frac{1}{b}$). Conversely, if $\|\boldsymbol{g}_{\boldsymbol{w}^*}^{\boldsymbol{D}_a}\|$ and $\mathrm{Tr}(\boldsymbol{H}_{\boldsymbol{w}^*}^{\boldsymbol{D}_a})$ are *small*, $\mathcal{G}(a)$ is *not that sensitive* to $s_{max}$ (or $b$). As a consequence, as the quantization bit-width $b$ decreases, the gap in $\mathcal{G}(a)$ values between different groups will further widen, leading to an increase in $\varphi(\boldsymbol{D})$

---

[2]Unless stated otherwise, $\|\cdot\|$ means $\ell_2$ norm in this paper.

[3]All proofs in this paper are presented in Appendix A.

Table 1. Fairness metric $\varphi(\boldsymbol{D})$ across various quantization methods and bit-widths for different models and datasets. A higher $\varphi(\boldsymbol{D})$ indicates more severe unfairness. Note: $\varphi(\boldsymbol{D})$ are presented as percentages due to their originally small magnitudes for improved clarity.

| Quantization Method | Bit-width | $\boldsymbol{\varphi(\boldsymbol{D})}$ (%) | | | | | |
| --- | --- | --- | --- | --- | --- | --- | --- |
| | | ResNet50 | | VGG19 | | ResNet18 | |
| | | UTK-Face | FER2013 | UTK-Face | FER2013 | CIFAR-10 | MNIST |
| PTQ | int 16 | 2.4 | 2.9 | 11.2 | 0.6 | 0.9 | 0.8 |
| | int 8 | 3.6 | 6.9 | 12.4 | 1.5 | 1.0 | 1.1 |
| | int 4 | 18.8 | 13.4 | 54.6 | 50.7 | 1.3 | 1.2 |
| QAT | int 16 | 3.3 | 3.6 | 11.7 | 1.4 | 1.1 | 1.0 |
| | int 8 | 6.0 | 9.9 | 18.9 | 2.9 | 1.3 | 1.1 |
| | int 4 | 25.4 | 44.4 | 84.1 | 65.7 | 1.5 | 1.4 |

and consequently exacerbating unfairness. The results are consistent with those shown in Table 1.

Our experimental results echo the arguments above. [4] We set up experiments using ResNet50 on the UTK-Face dataset for an ethnicity classification task. Experiments in Figure 1(a) and Figure 2 are towards the relationship between the gradient norm, the trace of the Hessian matrix, and the model accuracy, for five demographic attributes and their corresponding subgroups, at three different bit-widths in PTQ. Figure 1(a) and Figure 2 illustrate Theorem 4.1 in the following way [5]: (1) Consider a specific bit-width, for example, int 8. While the gradient norms and the traces of the Hessian matrices are *larger* over different groups, e.g., comparing group White and Indian, there is an *opposite* numerical relationship in accuracy; (2) Consider a specific demographic group, for example, group White. While its gradient norm accounts for the *smallest* percentage, its accuracy decreases the *most slowly*. Similar phenomena happen in traces of the Hessian matrices. Based on Theorem 4.1 and the observations from Figure 1(a) and Figure 2, we conclude our main findings as follows:

**Takeaway 1: In post-training quantization, gradient norms and traces of the Hessian matrices should pay for the exacerbation of unfairness. As bit-widths get lower, the accuracy after quantization on a demographic group with larger values of the two factors drops more dramatically.**

Taking a further step, we introduce that both $\|\boldsymbol{g}_{\boldsymbol{w}^*}^{\boldsymbol{D}_a}\|$ and $\text{Tr}(\boldsymbol{H}_{\boldsymbol{w}^*}^{\boldsymbol{D}_a})$ are negatively correlated to the size of the dataset

---

[4] We only present the accuracy results for training on UTK-Face dataset using ResNet50 in the main body of the paper for illustration, and leave similar experiments on other datasets and models to Appendix C.3.

[5] For a clear presentation, both gradient norms and traces of the Hessian matrices are scaled as a proportion of the sum among all demographic groups in this paper.

of subgroups, i.e., $|\boldsymbol{D}_a|$. Details will be discussed in the rest of this section.

**The effect of gradient norms.** The next lemma provides an upper bound for $\|\boldsymbol{g}_{\boldsymbol{w}^*}^{\boldsymbol{D}_{a_i}}\|$, revealing the factor contributing to the disparity in gradient norms among different groups.

**Lemma 4.2.** *For a given group $\boldsymbol{D}_{a_i}$ with a protected attribute $a_i \in \mathcal{A} = \{a_1, a_2, \cdots, a_k\}$, its gradient norm associated with the ERM function $\mathcal{L}$ evaluated at $\boldsymbol{w}^*$ can be upper bounded by:*

$$\|\boldsymbol{g}_{\boldsymbol{w}^*}^{\boldsymbol{D}_{a_i}}\| \leq \frac{1}{2|\boldsymbol{D}_{a_i}|} \sum_{j=1}^{k} |\boldsymbol{D}_{a_j}| \cdot \|\boldsymbol{g}_{\boldsymbol{w}^*}^{\boldsymbol{D}_{a_j}}\|, \quad (9)$$

*where $i \in [k]$ and $k = |\mathcal{A}| \geq 2$.*

**Corollary 4.3.** *Consider two groups $a$ and $b$ in $\mathcal{A}$. If $|\boldsymbol{D}_a| \leq |\boldsymbol{D}_b|$, then $\|\boldsymbol{g}_{\boldsymbol{w}^*}^{\boldsymbol{D}_a}\| \geq \|\boldsymbol{g}_{\boldsymbol{w}^*}^{\boldsymbol{D}_b}\|$.*

Lemma 4.2 associates the gradient norm of a specific group with its size, followed by a direct corollary when $\mathcal{A}$ only contains two subgroups. Figure 2 illustrates Corollary 4.3 by revealing the relationship between gradient norms $\|\boldsymbol{g}_{\boldsymbol{w}^*}^{\boldsymbol{D}_{a_i}}\|$ and sizes of groups $|\boldsymbol{D}_{a_i}|$ on the UTK-Face dataset. There exists a strong trend between decreasing group sizes and increasing gradient norms for such groups, i.e., $\|\boldsymbol{g}_{\boldsymbol{w}^*}^{\boldsymbol{D}_{a_i}}\| \propto \frac{1}{|\boldsymbol{D}_{a_i}|}$. This demonstrates that groups with *smaller* sizes of data points tend to have *larger* gradient norms than larger groups and vice-versa, leading to severer unfairness for underrepresented groups. The observation is actually not surprising because of an intuitive understanding: when a model converges at a local optimal, it learns little features from a disadvantaged group, inducing a low accuracy on this group. Now if we fine-tune the model individually on this disadvantaged group, the loss is large at the beginning and declines sharply, indicating a large norm of the corresponding gradient in this group.

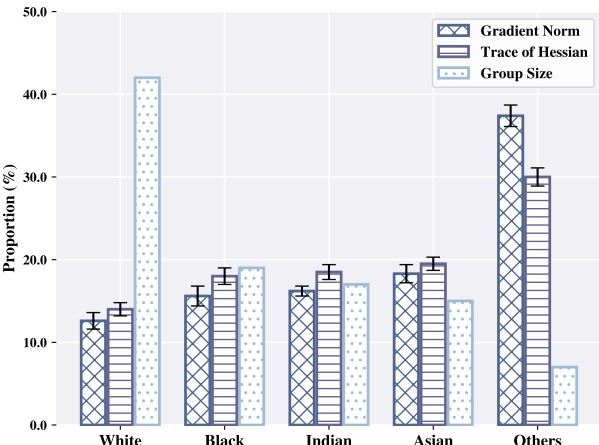

*Figure 2.* Proportions of gradient norms, traces of Hessian and group sizes for five demographic groups using the full-precision ResNet50 ($f_{\boldsymbol{w}^*}$) on the UTK-Face dataset in an ethnicity classification task.

**The effect of traces of the Hessian matrices.** A similar lemma establishes a connection between the trace of a group's Hessian and the size of the corresponding dataset.

**Lemma 4.4.** *For a given group $\boldsymbol{D}_{a_p}$ with a protected attribute $a_p \in \mathcal{A} = \{a_1, a_2, \cdots, a_k\}$, its trace of Hessian associated with the ERM function $\mathcal{L}$ evaluated at $\boldsymbol{w}^*$ can be upper bounded by:*

$$\mathrm{Tr}(\boldsymbol{H}_{\boldsymbol{w}^*}^{\boldsymbol{D}_{a_p}}) \leq \frac{1}{|\boldsymbol{D}_{a_p}|}\Big(n|\boldsymbol{D}|\lambda_{max}(\boldsymbol{H}_{\boldsymbol{w}^*}^{\boldsymbol{D}}) \\ - \sum_{j \neq p,q}^{k} |\boldsymbol{D}_{a_j}|\mathrm{Tr}(\boldsymbol{H}_{\boldsymbol{w}^*}^{\boldsymbol{D}_{a_j}})\Big), \quad (10)$$

*where $i \in [k]$, $k = |\mathcal{A}| \geq 2$ and $a_q$ is any attribute in $\mathcal{A}$ except $a_p$.*

Lemma 4.4 also indicates that the trace of the Hessian matrix for a specific group is related to the size of that group. Experiments in Figure 2 give evidence: groups with smaller sizes have larger traces of Hessians and vice-versa, i.e., $\mathrm{Tr}(\boldsymbol{H}_{\boldsymbol{w}^*}^{\boldsymbol{D}_{a_i}}) \propto \frac{1}{|\boldsymbol{D}_{a_i}|}$. This suggests that groups with fewer data points generally exhibit larger traces of Hessians compared to larger groups and vice-versa, resulting in more pronounced unfairness towards underrepresented groups.

**Experiments on a balanced dataset.** To further support our findings in Lemma 4.2 and Lemma 4.4, we experiment on a balanced dataset [6] CIFAR-10 (Krizhevsky et al., 2010)

---

[6] We also experiment on another balanced dataset MNIST with similar findings. Besides, for a clear comparison, we set up experiments on artificial Imbalanced-CIFAR-10 and Imbalanced-MNIST, whose training data is constructed by sampling different portions of images among different classes. Details and results are in Appendix C.1 and Appendix C.2.

with 10 groups and 10% of the total for each group. Figure 4(a) in Appendix C.1 reports our results and shows that as the bit-width gets lower in PTQ, accuracy gaps between 10 groups on the CIFAR-10 are relatively *stable*, while they grow *larger* on UTK-Face in Figure 1(a). As summarized in Table 1, the deterioration of unfairness on the two datasets is as follows: on CIFAR-10, the increase in $\varphi(\boldsymbol{D})$ is minimal, rising from 0.9% to 1.3%, whereas on the UTK-Face dataset, the increase in $\varphi(\boldsymbol{D})$ is much more substantial, rising from 2.4% to 18.8%.

**Takeaway 2: Class imbalance is to blame for unfairness, because of negative correlations between the group size and the gradient norm as well as the trace of Hessian.**

## 5. Fairness in Quantization-Aware Training

This section extends the analysis to the context of QAT. QAT models quantization during training and usually provides higher accuracy than PTQ schemes at lower bit-widths. The bound of excessive loss in QAT shares a large similarity to that in PTQ, but differs in two additional terms that are not negligible. To see details, let us consider one iteration $t+1$ of updating parameters using a widely-used optimizer, mini-batch SGD.

Recall that quantization stores the floating-point parameter vector $\boldsymbol{w}$ as a fixed-point $\boldsymbol{w}_{int}$ and computations are based on $\widetilde{\boldsymbol{w}}$. The update [7] is given by Krishnamoorthi (2018):

$$\boldsymbol{w}_{t+1} = \widetilde{\boldsymbol{w}}_t - \frac{\eta}{|\boldsymbol{B}|}\sum_{i=1}^{|\boldsymbol{B}|}\frac{\partial \ell}{\partial \widetilde{\boldsymbol{w}}_t}\Big(f_{\widetilde{\boldsymbol{w}}_t}(\boldsymbol{x}_i), y_i\Big), \quad (11)$$

where $\ell$ is assumed as a twice differentiable and convex loss function, $\eta$ is the learning rate, and $\boldsymbol{B}$ is the mini-batch. And $\boldsymbol{w}_t$ is approximated by $\widetilde{\boldsymbol{w}}_t$ after the quantization and de-quantization process, which is given by:

$$\widetilde{\boldsymbol{w}}_t = s \cdot \mathrm{clamp}\big(\lfloor \frac{\boldsymbol{w}_t}{s} \rceil; -2^{b-1}, 2^{b-1} - 1\big), \quad (12)$$

where $s$ is the scale inversely proportional to bit-width $b$. Further, recall that this induces an error $\Delta \boldsymbol{w}_t = \widetilde{\boldsymbol{w}}_t - \boldsymbol{w}_t$ upper bounded by $\frac{1}{2}\sqrt{n}s_{max}^2$.

The following result sheds light on the unfairness induced by QAT and provides a useful upper bound for a group's excessive loss.

**Theorem 5.1.** *Let $\ell$ be a twice differentiable loss function and consider a training process as defined in Eqn. (11) and a quantization process as defined in Eqn. (12). Then, the excessive loss gap for group $a \in \mathcal{A}$ at iteration $t+1$ is*

---

[7] The update relies on a straight-through estimator (STE) (Bengio et al., 2013), i.e. $\partial \widetilde{\boldsymbol{w}}_t / \partial \boldsymbol{w}_t = 1$. Details are in the proof of Theorem 5.1.

*upper bounded by:*

$$\mathcal{G}_{t+1}(a) \leq \underbrace{\frac{1}{2}\sqrt{n}s_{max}^2 \cdot \|\boldsymbol{g}_{\boldsymbol{w}_t}^{D_a}\| + \frac{1}{8}ns_{max}^4 \cdot \text{Tr}(\boldsymbol{H}_{\boldsymbol{w}_t}^{D_a})}_{\text{the upper bound of } \mathcal{G}(a) \text{ under PTQ}}$$

(13a)

$$+ \frac{1}{2}\sqrt{n}s_{max}^2 \cdot \|\boldsymbol{g}_{\boldsymbol{w}_t}^{D_a}\| \cdot \left(1 + 3\eta \cdot \text{Tr}(\boldsymbol{H}_{\boldsymbol{w}_t}^{D_a})\right)$$

$$+ \frac{1}{2}ns_{max}^4\eta \cdot \text{Tr}^2(\boldsymbol{H}_{\boldsymbol{w}_t}^{D_a}) + \mathcal{O}(\eta^2), \quad (13b)$$

*where* $\|\boldsymbol{g}_{\boldsymbol{w}_t}^{D_a}\|$ *and* $\text{Tr}(\boldsymbol{H}_{\boldsymbol{w}_t}^{D_a})$ *are defined in the same way as in Theorem 4.1.*

Although Eqn. (13a) and Eqn. (13b) give a larger bound than that in Theorem 4.1, they also shed light on a relation between fairness and the gradient norm $\|\boldsymbol{g}_{\boldsymbol{w}_t}^{D_a}\|$ and the trace of the Hessian $\text{Tr}(\boldsymbol{H}_{\boldsymbol{w}_t}^{D_a})$ in QAT. Similar theoretical conclusions can be carried over from Theorem 4.1, Lemma 4.2, and Lemma 4.4. We only present our empirical results here as a verification.

Figure 1(b) and Figure 2 show the relationship between the gradient norm, the trace of the Hessian matrix, and the model accuracy, for five demographic attributes and their corresponding subgroups, at three different bit-widths in QAT. Observations are quite similar to those from Figure 1(a) and Figure 2: (1) For a fixed bit-width, groups with *smaller* gradient norms and traces of Hessians tend to have *higher* accuracy and vice-versa; (2) The accuracy of each group decreases as the bit-width gets lower, however, the accuracy decreases *faster* for groups with *larger* gradient norms and traces of the Hessian matrices and vice-versa. In addition, we also conduct an experiment on CIFAR-10 in Figure 4(b), in which unfairness in accuracy is not worsened significantly. As presented in Table 1, the value of $\varphi(\boldsymbol{D})$ only increases slightly, from 1.1% to 1.5%.

**Comparison to PTQ: a larger gap in Eqn. (13b).** Recall that Theorem 5.1 decomposes the excessive loss $\mathcal{G}_{t+1}(a)$ at $t+1$-th iteration into two key components: the first term (13a) is identical to that in PTQ in Theorem 4.1, and the extra term (13b) relates together with gradient norms and traces of Hessian, which gives a larger bound than PTQ. To provide a clear comparison of this, we demonstrate in Table 1 on how *fairness metric* $\varphi(\boldsymbol{D})$ changes as bit-widths get lower. As shown, $\varphi(\boldsymbol{D})$ in QAT grow significantly *larger* than PTQ, echoing the difference in the theoretical bound. For instance, on the UTK-Face dataset with ResNet50, $\varphi(\boldsymbol{D})$ rises from 2.4% at int 16 to 18.8% at int 4 for PTQ, while for QAT, it increases from 3.3% to 25.4%. To provide more intuition, QAT exacerbates unfairness more than PTQ due to the dynamic interaction between gradient norms and Hessian traces under quantization constraints. Since QAT applies quantization throughout training, gradi-

ent updates must adapt to quantization-induced noise, leading to optimization in a more distorted loss landscape. In regions with high Hessian traces, the steep loss surface amplifies the effect of large gradient norms, causing uneven updates across subgroups. In contrast, PTQ quantizes only after full-precision training, avoiding these interaction effects and resulting in relatively lower unfairness.

**Takeaway 3: Although quantization-aware training always provides a better overall performance guarantee, deterioration in fairness induced by imbalanced datasets towards protected attributes is much more severe than that in post-training quantization.**

## 6. Mitigation Scheme and Evaluation

To further validate our main findings from theoretical and experimental analyses, that is, imbalanced datasets induce unfairness in model quantization, we adopt several data augmentation techniques as mitigation schemes. Data augmentation is viewed as a very powerful method to improve the generalizability of a deep model, especially in vision tasks, basically because the augmented data often provides a more comprehensive representation of data points, thus minimizing the distance between the training and validation set. In addition, data augmentation is also investigated as an effective way to alleviate class imbalance. We refer readers to a survey by Leevy et al. (2018) for details.

We consider two commonly used methods for image augmentations: *geometric transformations* and *random erasing* (Zhong et al., 2020) :

> **Geometric Transformations (GT).** For each class, randomly pick an image each time, and adopt one transformation from: (1) a rotation with a maximum degree of 20; (2) zooming with a factor between 1.1 and 1.2; (3) horizontal flipping, until enough images are generated.

> **Random Erasing (RE).** For each class, randomly pick an image to be augmented each time. *How to choose a mask of size* $n \times m$ is crucial in the effectiveness of such method. To simplify the process, we adopt a random strategy for selecting the mask size by enforcing a square mask, i.e., setting $n = m$. This is appropriate for datasets such as UTK-Face and FER2013, where images are uniformly sized at $48 \times 48$ pixels. During each augmentation step, a mask size parameter $n$ is randomly sampled from the set $\{3, 4, \ldots, 20\}$, derived from the optimal configuration in (Zhong et al., 2020). A patch of size $n \times n$ is then randomly selected from the image and masked with random values.

We evaluate the effectiveness of both methods by training ResNet-50 and VGG19 on UTK-Face and FER2013. We

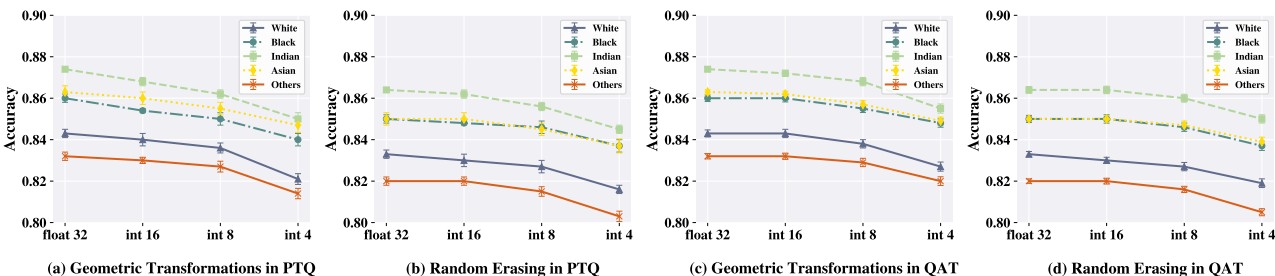

(a) Geometric Transformations in PTQ     (b) Random Erasing in PTQ     (c) Geometric Transformations in QAT     (d) Random Erasing in QAT

*Figure 3.* Mitigation schemes: Accuracy across five groups using ResNet50 on UTK-Face dataset by PTQ ((a) and (b)) and QAT ((c) and (d)). Both GT and RE are adopted on for data augmentation.

*Table 2.* Fairness metric $\varphi(\boldsymbol{D})$ for ResNet50 with data augmentation methods (GT and RE) applied.

| Augmentation Method | Quantization Method | $\varphi(\boldsymbol{D})$ (%) | | | | | |
|---|---|---|---|---|---|---|---|
| | | UTK-Face | | | FER2013 | | |
| | | int 16 | int 8 | int 4 | int 16 | int 8 | int 4 |
| **Non-Mitigation** | PTQ | 2.4 | 3.6 | 18.8 | 2.9 | 6.9 | 13.4 |
| | QAT | 3.3 | 6.0 | 25.4 | 3.6 | 9.9 | 44.4 |
| **GT** | PTQ | 0.8 | 1.1 | 2.6 | 1.8 | 2.3 | 2.9 |
| | QAT | 1.3 | 1.7 | 2.7 | 2.1 | 2.5 | 3.2 |
| **RE** | PTQ | 1.0 | 1.2 | 1.7 | 2.0 | 3.2 | 4.8 |
| | QAT | 1.1 | 1.3 | 2.0 | 2.8 | 3.9 | 4.9 |

present the results of training ResNet50 on UTK-Face using both PTQ and QAT, and leave other empirical evaluations to Appendix C.4. The UTK-Face dataset contains 18,964 images in training set, with 8,108 for White, 3,611 for Black, 3,176 for Indian, 2,718 for Asian, and 1,351 for Others. We balance them by augmenting training images until each containing 8,108 images.

Figure 3 presents the accuracy across five groups by PTQ ((a) and (b)) and QAT ((c) and (d)). Both GT and RE are adopted for data augmentation. Comparing to Figure 1, unfairness is significantly mitigated for both cases. For example, in QAT, the accuracy of group Others drops from 42% for 32-bit to 7% for 4-bit without data augmentation, while they drop from 82% and 83.2% for 32-bit to 80.3% and 82% for 4-bit when GT and RE are adopted, respectively. Furthermore, as shown in Table 2, the values of $\varphi(\boldsymbol{D})$ are significantly reduced after applying data augmentation methods, effectively mitigating unfairness.

Besides, as another evidence, our experiments on CIFAR-10 (and Imbalanced-CIFAR-10) and MNIST (and Imbalanced-MNIST) also support the effectiveness of data augmentation. See Appendix C.1 and Appendix C.2 for details.

We also compare the above random selection strategy of RE to some fixed choices of $n$, including $n = 3$, $n = 10$ and $n = 20$. The corresponding results are presented in Table 3.

As shown, whether under PTQ or QAT, fixing $n = 3$ or $n = 20$ is almost ineffective in mitigating unfairness. The fixed $n = 10$ strategy offers a slight improvement in fairness, but none of these fixed approaches are as effective as the random selection strategy for $n$. This suggests that, beyond the amount of augmented data, its quality also plays a crucial role in mitigating unfairness.

**Takeaway 4: Data augmentation helps mitigate the unfairness caused by quantization.**

## 7. Conclusion and Future Work

Starting from an observation in an experiment on quantization and accuracy across different subgroups of UTK-Face, we are aware of a significant concern for equality protection and privacy preservation. To the best of our knowledge, this is the first paper to both address this issue and go in-depth on its causes and mitigation theoretically and empirically. Our main findings are concluded in "Takeaways". For future work, we will validate our findings in other vision tasks, such as detection, or other areas where compressed deep models are needed, such as NLP. Besides, we aim to explore solutions beyond data augmentation to mitigate unfairness.

*Table 3.* Fairness metric $\varphi(\boldsymbol{D})$ for ResNet50 with **fixed** patch sizes $n$ over the range $\{3, 10, 20\}$, compared with the **random** selection strategy of $n$ from the range $\{3, 4, \ldots, 20\}$ (referred to as the Optimal Strategy) and no data augmentation (referred to as the Baseline), in random erasing.

| Patch Size | Quantization Method | $\varphi(\boldsymbol{D})$ (%) | | | | | |
|---|---|---|---|---|---|---|---|
| | | UTK-Face | | | FER2013 | | |
| | | int 16 | int 8 | int 4 | int 16 | int 8 | int 4 |
| **Baseline** | PTQ | **2.4** | **3.6** | **18.8** | **2.9** | **6.9** | **13.4** |
| | QAT | **3.3** | **6.0** | **25.4** | **3.6** | **9.9** | **44.4** |
| **Optimal Strategy** | PTQ | **1.0** | **1.2** | **1.7** | **2.0** | **3.2** | **4.8** |
| | QAT | **1.1** | **1.3** | **2.0** | **2.8** | **3.9** | **4.9** |
| $n = 3$ | PTQ | 2.2 | 3.3 | 18.8 | 2.5 | 6.3 | 12.1 |
| | QAT | 3.1 | 5.8 | 25.0 | 3.2 | 8.7 | 40.3 |
| $n = 10$ | PTQ | 1.4 | 2.1 | 10.6 | 2.3 | 4.3 | 8.7 |
| | QAT | 1.5 | 2.4 | 11.9 | 3.0 | 5.6 | 10.5 |
| $n = 20$ | PTQ | 2.3 | 3.6 | 18.7 | 2.7 | 6.5 | 12.8 |
| | QAT | 3.3 | 5.9 | 25.2 | 3.3 | 9.1 | 41.9 |

## Acknowledgements

Qiankun Zhang is supported by the National Natural Science Foundation of China (Grant 62302183), Open Foundation of Key Laboratory of Cyberspace Security, Ministry of Education of China (Grant KLCS20240401), Ant Group Research Fund (Grant 20242452) and CCF-DiDi GAIA Collaborative Research Funds (Grant CCF-DiDi GAIA 202412). Jing Wang is supported by the National Natural Science Foundation of China (Grant 62202197), and the Major Research Project of Hubei Province (Grant 2023BAA027). Bin Yuan is supported by the National Natural Science Foundation of China (Grant 62372191), the Open Topics from The Lion Rock Labs of Cyberspace Security (Grant LRL24013), and Songshan Laboratory (Grant 241110210200). Xianjun Deng is supported by the National Key R&D Program of China (Grant 2022YFE0138600), and the National Natural Science Foundation of China (Grant U24B20153).

## Impact Statement

This paper presents work whose goal is to advance the field of Machine Learning. There are many potential societal consequences of our work, none which we feel must be specifically highlighted here.

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

# A. Missing Proofs

## A.1. Proof of Eqn. (4)

*Proof.* Let $w = (w_1, w_2, \cdots, w_i, \cdots, w_n)$ be an $n$-dimensional vector whose component $w_i$ is a real number. $\tilde{w}$ is the approximate vector of $w$ obtained after $w$ has been quantized and de-quantized.

Since the distribution of the parameters $w \in [w_{min}, w_{max}]$ of the neural network is usually symmetric about 0, i.e., $w_{min} = -w_{max}$, selecting a scaling factor $s = \frac{w_{max}}{2^{b-1}-1}$ ensures that the quantization Eqn. (3) does not incur clipping errors. The reason is as follows:

$$\frac{w_{max}}{s} = 2^{b-1} - 1 \,, \frac{w_{min}}{s} = -2^{b-1} + 1 \,,$$

$$-2^{b-1} < -2^{b-1} + 1 \leq \lfloor \frac{w}{s} \rceil \leq 2^{b-1} - 1 \,.$$

Thus, the quantization error of this method comes only from the $\lfloor \cdot \rceil$ operator, i.e.:

$$|\Delta w_i| = |\tilde{w}_i - w_i| \leq s_i \cdot \frac{1}{2} s_i = \frac{1}{2} s_i^2 \,,$$

where $s_i$ in the inequality comes from the coefficient of the de-quantization formula Eqn. (2) ($0 < s_i < 1$), $\frac{1}{2} s_i$ comes from the error of the $\lfloor \cdot \rceil$ operator.

For the $\ell_2$ norm of the error vector $\Delta w$:

$$\|\Delta w\| = \|(\Delta w_1, \Delta w_2, \cdots, \Delta w_i, \cdots, \Delta w_n)\|$$

$$= \sqrt{\sum_{i=1}^{n} \Delta w_i^2} \leq \sqrt{\sum_{i=1}^{n} \frac{1}{4} s_i^4} \leq \sqrt{\frac{1}{4} n s_{max}^4} = \frac{1}{2} \sqrt{n} s_{max}^2 \,,$$

where $s_{max}$ is the largest scale used in the quantization among all parameters. $\qquad \square$

## A.2. Proof of Theorem 4.1

*Proof.* First recall that we assume the loss function $\ell(\cdot)$ is twice differentiable. We use a second-order Taylor expansion around $w^*$. The ERM function $\mathcal{L}(\tilde{w}^*; D_a)$ for a group $a \in \mathcal{A}$ can be stated as:

$$\mathcal{L}(\tilde{w}^*; D_a) = \mathcal{L}(w^*; D_a) + (\Delta w^*)^{\mathrm{T}} g_{w^*}^{D_a} + \frac{1}{2} (\Delta w^*)^{\mathrm{T}} H_{w^*}^{D_a} (\Delta w^*) + \mathcal{O}\left(\|\Delta w^*\|^3\right) \,.$$

The excessive loss $\mathcal{G}(a)$ for a group $a \in \mathcal{A}$ is then given by definition:

$$\mathcal{G}(a) = \mathcal{L}(\tilde{w}^*; D_a) - \mathcal{L}(w^*; D_a)$$

$$= (\Delta w^*)^{\mathrm{T}} g_{w^*}^{D_a} + \frac{1}{2} (\Delta w^*)^{\mathrm{T}} H_{w^*}^{D_a} (\Delta w^*) + \mathcal{O}\left(\|\Delta w^*\|^3\right) \,.$$

Further recall in Eqn. (4), the quantizaton error $\|\Delta w^*\| = \|\tilde{w}^* - w^*\|$ is upper bounded by:

$$\|\Delta w^*\| \leq \frac{1}{2} \sqrt{n} s_{max}^2 \,.$$

Combining with a Cauchy-Schwarz inequality, it follows by:

$$(\Delta w^*)^{\mathrm{T}} g_{w^*}^{D_a} \leq \|\Delta w^*\| \cdot \|g_{w^*}^{D_a}\| \leq \frac{1}{2} \sqrt{n} s_{max}^2 \cdot \|g_{w^*}^{D_a}\| \,. \tag{14}$$

For the second-order derivative term, combining Eqn. (4) and the consistency between the matrix $\ell_2$ norm and the vector $\ell_2$ norm, we have:

$$\frac{1}{2} (\Delta w^*)^{\mathrm{T}} H_{w^*}^{D_a} (\Delta w^*) \leq \frac{1}{2} \|\Delta w^*\|^2 \cdot \|H_{w^*}^{D_a}\| \leq \frac{1}{8} n s_{max}^4 \cdot \|H_{w^*}^{D_a}\| \,.$$

Besides, since $\boldsymbol{H}_{\boldsymbol{w}^*}^{D_a}$ is a real symmetric matrix, $\|\boldsymbol{H}_{\boldsymbol{w}^*}^{D_a}\| = \max_i |\lambda_i|$, where $\lambda_i$ is the eigenvalue of $\boldsymbol{H}_{\boldsymbol{w}^*}^{D_a}$. Moreover, the matrix $\boldsymbol{H}_{\boldsymbol{w}^*}^{D_a}$ is positive semi-definite, meaning all eigenvalues of the matrix $\boldsymbol{H}_{\boldsymbol{w}^*}^{D_a}$ are greater than or equal to $0$. This property holds for convex loss functions and also for non-convex ones, as the second-order Taylor expansion at a local optimum approximates the loss as convex (Nocedal & Wright, 1999). So $\|\boldsymbol{H}_{\boldsymbol{w}^*}^{D_a}\| = \max_i \lambda_i$. According to the property of the trace of the matrix that the trace of a matrix is equal to the sum of all eigenvalues of the matrix, we have:

$$\|\boldsymbol{H}_{\boldsymbol{w}^*}^{D_a}\| = \max_i \lambda_i \leq \sum_i \lambda_i = \mathrm{Tr}(\boldsymbol{H}_{\boldsymbol{w}^*}^{D_a}) \ .$$

Thus,

$$\frac{1}{2}(\Delta \boldsymbol{w}^*)^{\mathrm{T}} \boldsymbol{H}_{\boldsymbol{w}^*}^{D_a} (\Delta \boldsymbol{w}^*) \leq \frac{1}{8} n s_{max}^4 \cdot \mathrm{Tr}(\boldsymbol{H}_{\boldsymbol{w}^*}^{D_a}) \ . \tag{15}$$

The upper bound for the excessive loss $\mathcal{G}(a)$ is thus obtained by combining Eqn. (14) and Eqn. (15). $\qquad\square$

### A.3. Proof of Lemma 4.2

*Proof.* By the assumption that the model converges to a local minimum, it follows that:

$$
\begin{aligned}
\boldsymbol{g}_{\boldsymbol{w}^*}^{D} &= \nabla_{\boldsymbol{w}} \mathcal{L}(\boldsymbol{w}^*; \boldsymbol{D}) \\
&= \frac{1}{|\boldsymbol{D}|} \sum_{i=1}^{|\boldsymbol{D}|} \boldsymbol{g}_{\boldsymbol{w}^*}^{(\boldsymbol{x}_i, y_i)} \\
&= \frac{1}{|\boldsymbol{D}|} \sum_{j=1}^{k} \sum_{i=1}^{|\boldsymbol{D}_{a_j}|} \boldsymbol{g}_{\boldsymbol{w}^*}^{(\boldsymbol{x}_i, y_i)} \\
&= \sum_{j=1}^{k} \left( \frac{|\boldsymbol{D}_{a_j}|}{|\boldsymbol{D}|} \frac{1}{|\boldsymbol{D}_{a_j}|} \sum_{i=1}^{|\boldsymbol{D}_{a_j}|} \boldsymbol{g}_{\boldsymbol{w}^*}^{(\boldsymbol{x}_i, y_i)} \right) \\
&= \sum_{j=1}^{k} \frac{|\boldsymbol{D}_{a_j}|}{|\boldsymbol{D}|} \boldsymbol{g}_{\boldsymbol{w}^*}^{D_{a_j}} \\
&= 0 \ .
\end{aligned}
\tag{16}
$$

Thus, for group $a_i \in \mathcal{A}$, we have:

$$\boldsymbol{g}_{\boldsymbol{w}^*}^{D_{a_i}} = -\frac{1}{|\boldsymbol{D}_{a_i}|} \sum_{j \neq i}^{k} |\boldsymbol{D}_{a_j}| \boldsymbol{g}_{\boldsymbol{w}^*}^{D_{a_j}} \ .$$

Besides, by the trigonometric inequality property of the vector norm, it follows that:

$$\|\boldsymbol{g}_{\boldsymbol{w}^*}^{D_{a_i}}\| \leq \frac{1}{|\boldsymbol{D}_{a_i}|} \sum_{j \neq i}^{k} |\boldsymbol{D}_{a_j}| \cdot \|\boldsymbol{g}_{\boldsymbol{w}^*}^{D_{a_j}}\| \ . \tag{17}$$

Add $\|\boldsymbol{g}_{\boldsymbol{w}^*}^{D_{a_i}}\|$ to both sides of Eqn.( 17):

$$2\|\boldsymbol{g}_{\boldsymbol{w}^*}^{D_{a_i}}\| \leq \frac{1}{|\boldsymbol{D}_{a_i}|} \sum_{j=1}^{k} |\boldsymbol{D}_{a_j}| \cdot \|\boldsymbol{g}_{\boldsymbol{w}^*}^{D_{a_j}}\| \ . \tag{18}$$

Divide both sides of Eqn. (18) by 2:

$$\|\boldsymbol{g}_{\boldsymbol{w}^*}^{D_{a_i}}\| \leq \frac{1}{2|\boldsymbol{D}_{a_i}|} \sum_{j=1}^{k} |\boldsymbol{D}_{a_j}| \cdot \|\boldsymbol{g}_{\boldsymbol{w}^*}^{D_{a_j}}\| \ .$$

$\qquad\square$

## A.4. Proof of Corollary 4.3

*Proof.* For two groups $a$ and $b$ in $\mathcal{A}$, by the assumption that the model converges to a local minimum, it follows that:

$$
\begin{aligned}
\boldsymbol{g}_{\boldsymbol{w}^*}^{\boldsymbol{D}} &= \nabla_{\boldsymbol{w}} \mathcal{L}(\boldsymbol{w}^*; \boldsymbol{D}) \\
&= \frac{1}{|\boldsymbol{D}|} \sum_{i=1}^{|\boldsymbol{D}|} \boldsymbol{g}_{\boldsymbol{w}^*}^{(\boldsymbol{x}_i, y_i)} \\
&= \frac{1}{|\boldsymbol{D}|} \sum_{i=1}^{|\boldsymbol{D}_a|} \boldsymbol{g}_{\boldsymbol{w}^*}^{(\boldsymbol{x}_i, y_i)} + \frac{1}{|\boldsymbol{D}|} \sum_{i=1}^{|\boldsymbol{D}_b|} \boldsymbol{g}_{\boldsymbol{w}^*}^{(\boldsymbol{x}_i, y_i)} \\
&= \frac{|\boldsymbol{D}_a|}{|\boldsymbol{D}|} \frac{1}{|\boldsymbol{D}_a|} \sum_{i=1}^{|\boldsymbol{D}_a|} \boldsymbol{g}_{\boldsymbol{w}^*}^{(\boldsymbol{x}_i, y_i)} + \frac{|\boldsymbol{D}_b|}{|\boldsymbol{D}|} \frac{1}{|\boldsymbol{D}_b|} \sum_{i=1}^{|\boldsymbol{D}_b|} \boldsymbol{g}_{\boldsymbol{w}^*}^{(\boldsymbol{x}_i, y_i)} \\
&= \frac{|\boldsymbol{D}_a|}{|\boldsymbol{D}|} \boldsymbol{g}_{\boldsymbol{w}^*}^{\boldsymbol{D}_a} + \frac{|\boldsymbol{D}_b|}{|\boldsymbol{D}|} \boldsymbol{g}_{\boldsymbol{w}^*}^{\boldsymbol{D}_b} \\
&= 0 \, .
\end{aligned}
$$

Thus, we have:

$$
\boldsymbol{g}_{\boldsymbol{w}^*}^{\boldsymbol{D}_a} = -\frac{|\boldsymbol{D}_b|}{|\boldsymbol{D}_a|} \boldsymbol{g}_{\boldsymbol{w}^*}^{\boldsymbol{D}_b} \, , \quad \|\boldsymbol{g}_{\boldsymbol{w}^*}^{\boldsymbol{D}_a}\| = \frac{|\boldsymbol{D}_b|}{|\boldsymbol{D}_a|} \|\boldsymbol{g}_{\boldsymbol{w}^*}^{\boldsymbol{D}_b}\| \, .
$$

If $|\boldsymbol{D}_a| \leq |\boldsymbol{D}_b|$, then $\|\boldsymbol{g}_{\boldsymbol{w}^*}^{\boldsymbol{D}_a}\| \geq \|\boldsymbol{g}_{\boldsymbol{w}^*}^{\boldsymbol{D}_b}\|$. $\qquad\square$

## A.5. Proof of Lemma 4.4

*Proof.* In the same way as the derivation of $\boldsymbol{g}_{\boldsymbol{w}^*}^{\boldsymbol{D}}$ in Eqn. (16), we have:

$$
\boldsymbol{H}_{\boldsymbol{w}^*}^{\boldsymbol{D}} = \sum_{j=1}^{k} \frac{|\boldsymbol{D}_{a_j}|}{|\boldsymbol{D}|} \boldsymbol{H}_{\boldsymbol{w}^*}^{\boldsymbol{D}_{a_j}} \, .
$$

By the simple property of the trace of the matrix and the relationship between the trace and eigenvalues of the matrix, it follows that:

$$
\mathrm{Tr}(\boldsymbol{H}_{\boldsymbol{w}^*}^{\boldsymbol{D}}) = \mathrm{Tr}\Big( \sum_{j=1}^{k} \frac{|\boldsymbol{D}_{a_j}|}{|\boldsymbol{D}|} \boldsymbol{H}_{\boldsymbol{w}^*}^{\boldsymbol{D}_{a_j}} \Big) = \sum_{j=1}^{k} \frac{|\boldsymbol{D}_{a_j}|}{|\boldsymbol{D}|} \mathrm{Tr}(\boldsymbol{H}_{\boldsymbol{w}^*}^{\boldsymbol{D}_{a_j}}) \leq n \lambda_{max}(\boldsymbol{H}_{\boldsymbol{w}^*}^{\boldsymbol{D}})
$$

For group $a_p$, we have:

$$
\begin{aligned}
\frac{|\boldsymbol{D}_{a_p}|}{|\boldsymbol{D}|} \mathrm{Tr}(\boldsymbol{H}_{\boldsymbol{w}^*}^{\boldsymbol{D}_{a_p}}) &\leq n \lambda_{max}(\boldsymbol{H}_{\boldsymbol{w}^*}^{\boldsymbol{D}}) - \sum_{j \neq p}^{k} \frac{|\boldsymbol{D}_{a_j}|}{|\boldsymbol{D}|} \mathrm{Tr}(\boldsymbol{H}_{\boldsymbol{w}^*}^{\boldsymbol{D}_{a_j}}) \\
&\leq n \lambda_{max}(\boldsymbol{H}_{\boldsymbol{w}^*}^{\boldsymbol{D}}) - \sum_{j \neq p,q}^{k} \frac{|\boldsymbol{D}_{a_j}|}{|\boldsymbol{D}|} \mathrm{Tr}(\boldsymbol{H}_{\boldsymbol{w}^*}^{\boldsymbol{D}_{a_j}}) \, .
\end{aligned}
$$

Thus,

$$
\mathrm{Tr}(\boldsymbol{H}_{\boldsymbol{w}^*}^{\boldsymbol{D}_{a_p}}) \leq \frac{1}{|\boldsymbol{D}_{a_p}|} \Big( n|\boldsymbol{D}| \lambda_{max}(\boldsymbol{H}_{\boldsymbol{w}^*}^{\boldsymbol{D}}) - \sum_{j \neq p,q}^{k} |\boldsymbol{D}_{a_j}| \mathrm{Tr}(\boldsymbol{H}_{\boldsymbol{w}^*}^{\boldsymbol{D}_{a_j}}) \Big) \, .
$$

$\qquad\square$

## A.6. Proof of Theorem 5.1

*Proof.* The proof of Theorem 5.1 relies on the following two second order Taylor approximations: (1) The first approximates the ERM function at iteration $t+1$ under non-quantized training, i.e., $\boldsymbol{w}_{t+1} = \boldsymbol{w}_t - \eta \boldsymbol{g}_{\boldsymbol{w}_t}^{B}$, where $\boldsymbol{B} \in \boldsymbol{D}$ denotes the

mini-batch. (2) The second approximates the ERM function under quantized training, where the parameter vector is updated by:

$$\boldsymbol{w}_{t+1} = \widetilde{\boldsymbol{w}}_t - \frac{\eta}{|\boldsymbol{B}|} \sum_{i=1}^{|\boldsymbol{B}|} \frac{\partial \ell}{\partial \boldsymbol{w}_t}\left(f_{\widetilde{\boldsymbol{w}}_t}(\boldsymbol{x}_i), y_i\right)$$

$$= \widetilde{\boldsymbol{w}}_t - \frac{\eta}{|\boldsymbol{B}|} \sum_{i=1}^{|\boldsymbol{B}|} \frac{\partial \ell}{\partial \widetilde{\boldsymbol{w}}_t}\left(f_{\widetilde{\boldsymbol{w}}_t}(\boldsymbol{x}_i), y_i\right) \cdot \frac{\partial \widetilde{\boldsymbol{w}}_t}{\partial \boldsymbol{w}_t}$$

$$= \widetilde{\boldsymbol{w}}_t - \frac{\eta}{|\boldsymbol{B}|} \sum_{i=1}^{|\boldsymbol{B}|} \frac{\partial \ell}{\partial \widetilde{\boldsymbol{w}}_t}\left(f_{\widetilde{\boldsymbol{w}}_t}(\boldsymbol{x}_i), y_i\right)$$

$$= \widetilde{\boldsymbol{w}}_t - \eta \boldsymbol{g}_{\widetilde{\boldsymbol{w}}_t}^{\boldsymbol{B}} ,$$

where $\frac{\partial \widetilde{\boldsymbol{w}}_t}{\partial \boldsymbol{w}_t} = 1$ by the straight-through estimator (STE) (Bengio et al., 2013; Nagel et al., 2021). Finally, the result is obtained by taking the difference of these approximations under quantized and non-quantized training.

**1. Approximation of non-quantized ERM.** The approximation of non-quantized ERM can be derived using a second order Taylor approximation as follows:

$$\mathcal{L}(\boldsymbol{w}_{t+1}; \boldsymbol{D}_a) = \mathcal{L}(\boldsymbol{w}_t - \eta \boldsymbol{g}_{\boldsymbol{w}_t}^{\boldsymbol{B}}; \boldsymbol{D}_a) \approx \mathcal{L}(\boldsymbol{w}_t; \boldsymbol{D}_a) - \eta (\boldsymbol{g}_{\boldsymbol{w}_t}^{\boldsymbol{B}})^{\mathrm{T}} \boldsymbol{g}_{\boldsymbol{w}_t}^{\boldsymbol{D}_a} + \frac{1}{2}\eta^2 (\boldsymbol{g}_{\boldsymbol{w}_t}^{\boldsymbol{B}})^{\mathrm{T}} \boldsymbol{H}_{\boldsymbol{w}_t}^{\boldsymbol{D}_a} \boldsymbol{g}_{\boldsymbol{w}_t}^{\boldsymbol{B}} . \tag{19}$$

**2. Approximation of quantized ERM.** Consider one iteration $t + 1$ of updating parameters using the mini-batch SGD, i.e., $\boldsymbol{w}_{t+1} = \widetilde{\boldsymbol{w}}_t - \eta \boldsymbol{g}_{\widetilde{\boldsymbol{w}}_t}^{\boldsymbol{B}}$. Recall that quantization stores the float-point parameter vector $\boldsymbol{w}$ as a fixed-point $\boldsymbol{w}_{int}$ and computation is based on $\widetilde{\boldsymbol{w}}$. $\boldsymbol{w}_{t+1}$ is approximated by $\widetilde{\boldsymbol{w}}_{t+1}$ when participating in the next round of iterations of the computation. Hence,

$$\widetilde{\boldsymbol{w}}_{t+1} = \boldsymbol{w}_{t+1} + \Delta \boldsymbol{w}_{t+1} = \widetilde{\boldsymbol{w}}_t - \eta \boldsymbol{g}_{\widetilde{\boldsymbol{w}}_t}^{\boldsymbol{B}} + \Delta \boldsymbol{w}_{t+1} = \boldsymbol{w}_t + \Delta \boldsymbol{w}_t - \eta \boldsymbol{g}_{\widetilde{\boldsymbol{w}}_t}^{\boldsymbol{B}} + \Delta \boldsymbol{w}_{t+1} .$$

Applying a second order Taylor approximation around $\boldsymbol{w}_t$ allows us to estimate the quantized ERM function at iteration t + 1 as:

$$\mathcal{L}(\widetilde{\boldsymbol{w}}_{t+1}; \boldsymbol{D}_a)$$

$$= \mathcal{L}(\boldsymbol{w}_t + \Delta \boldsymbol{w}_t + \Delta \boldsymbol{w}_{t+1} - \eta \boldsymbol{g}_{\widetilde{\boldsymbol{w}}_t}^{\boldsymbol{B}}; \boldsymbol{D}_a) \tag{20a}$$

$$\approx \mathcal{L}(\boldsymbol{w}_t; \boldsymbol{D}_a) + (\Delta \boldsymbol{w}_t + \Delta \boldsymbol{w}_{t+1} - \eta \boldsymbol{g}_{\widetilde{\boldsymbol{w}}_t}^{\boldsymbol{B}})^{\mathrm{T}} \boldsymbol{g}_{\boldsymbol{w}_t}^{\boldsymbol{D}_a} \tag{20b}$$

$$+ \frac{1}{2}(\Delta \boldsymbol{w}_t + \Delta \boldsymbol{w}_{t+1} - \eta \boldsymbol{g}_{\widetilde{\boldsymbol{w}}_t}^{\boldsymbol{B}})^{\mathrm{T}} \cdot \boldsymbol{H}_{\boldsymbol{w}_t}^{\boldsymbol{D}_a}(\Delta \boldsymbol{w}_t + \Delta \boldsymbol{w}_{t+1} - \eta \boldsymbol{g}_{\widetilde{\boldsymbol{w}}_t}^{\boldsymbol{B}})$$

$$\approx \mathcal{L}(\boldsymbol{w}_t; \boldsymbol{D}_a) + (\Delta \boldsymbol{w}_t + \Delta \boldsymbol{w}_{t+1} - \eta \boldsymbol{g}_{\boldsymbol{w}_t}^{\boldsymbol{B}} - \eta \boldsymbol{H}_{\boldsymbol{w}_t}^{\boldsymbol{B}} \Delta \boldsymbol{w}_t)^{\mathrm{T}} \boldsymbol{g}_{\boldsymbol{w}_t}^{\boldsymbol{D}_a} \tag{20c}$$

$$+ \frac{1}{2}(\Delta \boldsymbol{w}_t + \Delta \boldsymbol{w}_{t+1} - \eta \boldsymbol{g}_{\boldsymbol{w}_t}^{\boldsymbol{B}} - \eta \boldsymbol{H}_{\boldsymbol{w}_t}^{\boldsymbol{B}} \Delta \boldsymbol{w}_t)^{\mathrm{T}} \boldsymbol{H}_{\boldsymbol{w}_t}^{\boldsymbol{D}_a} \cdot (\Delta \boldsymbol{w}_t + \Delta \boldsymbol{w}_{t+1} - \eta \boldsymbol{g}_{\boldsymbol{w}_t}^{\boldsymbol{B}} - \eta \boldsymbol{H}_{\boldsymbol{w}_t}^{\boldsymbol{B}} \Delta \boldsymbol{w}_t)$$

$$= \mathcal{L}(\boldsymbol{w}_t; \boldsymbol{D}_a) + (\Delta \boldsymbol{w}_t + \Delta \boldsymbol{w}_{t+1})^{\mathrm{T}} \boldsymbol{g}_{\boldsymbol{w}_t}^{\boldsymbol{D}_a} - \eta (\boldsymbol{g}_{\boldsymbol{w}_t}^{\boldsymbol{B}})^{\mathrm{T}} \boldsymbol{g}_{\boldsymbol{w}_t}^{\boldsymbol{D}_a} - \eta (\boldsymbol{H}_{\boldsymbol{w}_t}^{\boldsymbol{B}} \Delta \boldsymbol{w}_t)^{\mathrm{T}} \boldsymbol{g}_{\boldsymbol{w}_t}^{\boldsymbol{D}_a} \tag{20d}$$

$$+ \frac{1}{2}(\Delta \boldsymbol{w}_t + \Delta \boldsymbol{w}_{t+1})^{\mathrm{T}} \boldsymbol{H}_{\boldsymbol{w}_t}^{\boldsymbol{D}_a}(\Delta \boldsymbol{w}_t + \Delta \boldsymbol{w}_{t+1}) + \frac{1}{2}\eta^2 (\boldsymbol{g}_{\boldsymbol{w}_t}^{\boldsymbol{B}})^{\mathrm{T}} \boldsymbol{H}_{\boldsymbol{w}_t}^{\boldsymbol{D}_a} \boldsymbol{g}_{\boldsymbol{w}_t}^{\boldsymbol{B}}$$

$$- \eta (\Delta \boldsymbol{w}_t + \Delta \boldsymbol{w}_{t+1})^{\mathrm{T}} \boldsymbol{H}_{\boldsymbol{w}_t}^{\boldsymbol{D}_a} \boldsymbol{g}_{\boldsymbol{w}_t}^{\boldsymbol{B}} - \eta (\Delta \boldsymbol{w}_t + \Delta \boldsymbol{w}_{t+1})^{\mathrm{T}} \boldsymbol{H}_{\boldsymbol{w}_t}^{\boldsymbol{D}_a}(\boldsymbol{H}_{\boldsymbol{w}_t}^{\boldsymbol{B}} \Delta \boldsymbol{w}_t) + \mathcal{O}(\eta^2) ,$$

where Eqn. (20c) follows from the first-order Taylor expansion approximation of $\boldsymbol{g}_{\widetilde{\boldsymbol{w}}_t}^{\boldsymbol{B}}$ at $\boldsymbol{w}_t$:

$$\boldsymbol{g}_{\widetilde{\boldsymbol{w}}_t}^{\boldsymbol{B}} \approx \boldsymbol{g}_{\boldsymbol{w}_t}^{\boldsymbol{B}} + \Delta \boldsymbol{w}_t \boldsymbol{H}_{\boldsymbol{w}_t}^{\boldsymbol{D}_a} .$$

The upper bound for the excessive loss $\mathcal{G}_{t+1}(a)$ can be obtained by combining Eqn. (19) and Eqn. (20d):

$$\mathcal{G}_{t+1}(a)$$

$$=\mathcal{L}(\widetilde{\boldsymbol{w}}_{t+1}; \boldsymbol{D}_a) - \mathcal{L}(\boldsymbol{w}_{t+1}; \boldsymbol{D}_a) \tag{21a}$$

$$\approx (\Delta\boldsymbol{w}_t + \Delta\boldsymbol{w}_{t+1})^{\mathrm{T}} \boldsymbol{g}_{\boldsymbol{w}_t}^{\boldsymbol{D}_a} - \eta(\boldsymbol{H}_{\boldsymbol{w}_t}^{\boldsymbol{B}} \Delta\boldsymbol{w}_t)^{\mathrm{T}} \boldsymbol{g}_{\boldsymbol{w}_t}^{\boldsymbol{D}_a} + \frac{1}{2}(\Delta\boldsymbol{w}_t + \Delta\boldsymbol{w}_{t+1})^{\mathrm{T}} \boldsymbol{H}_{\boldsymbol{w}_t}^{\boldsymbol{D}_a}(\Delta\boldsymbol{w}_t + \Delta\boldsymbol{w}_{t+1}) \tag{21b}$$

$$- \eta(\Delta\boldsymbol{w}_t + \Delta\boldsymbol{w}_{t+1})^{\mathrm{T}} \boldsymbol{H}_{\boldsymbol{w}_t}^{\boldsymbol{D}_a} \boldsymbol{g}_{\boldsymbol{w}_t}^{\boldsymbol{B}} - \eta(\Delta\boldsymbol{w}_t + \Delta\boldsymbol{w}_{t+1})^{\mathrm{T}} \boldsymbol{H}_{\boldsymbol{w}_t}^{\boldsymbol{D}_a}(\boldsymbol{H}_{\boldsymbol{w}_t}^{\boldsymbol{B}} \Delta\boldsymbol{w}_t) + \mathcal{O}(\eta^2)$$

$$\leq \|\Delta\boldsymbol{w}_t + \Delta\boldsymbol{w}_{t+1}\| \cdot \|\boldsymbol{g}_{\boldsymbol{w}_t}^{\boldsymbol{D}_a}\| + \eta\|\Delta\boldsymbol{w}_t\| \cdot \|\boldsymbol{H}_{\boldsymbol{w}_t}^{\boldsymbol{B}}\| \cdot \|\boldsymbol{g}_{\boldsymbol{w}_t}^{\boldsymbol{D}_a}\| + \frac{1}{2}\|\Delta\boldsymbol{w}_t + \Delta\boldsymbol{w}_{t+1}\|^2 \cdot \|\boldsymbol{H}_{\boldsymbol{w}_t}^{\boldsymbol{D}_a}\| \tag{21c}$$

$$+ \eta\|\Delta\boldsymbol{w}_t + \Delta\boldsymbol{w}_{t+1}\| \cdot \|\boldsymbol{H}_{\boldsymbol{w}_t}^{\boldsymbol{D}_a}\| \cdot \|\boldsymbol{g}_{\boldsymbol{w}_t}^{\boldsymbol{B}}\| + \eta\|\Delta\boldsymbol{w}_t + \Delta\boldsymbol{w}_{t+1}\| \cdot \|\boldsymbol{H}_{\boldsymbol{w}_t}^{\boldsymbol{D}_a}\| \cdot \|\boldsymbol{H}_{\boldsymbol{w}_t}^{\boldsymbol{B}}\| \cdot \|\Delta\boldsymbol{w}_t\| + \mathcal{O}(\eta^2)$$

$$\approx \|\Delta\boldsymbol{w}_t + \Delta\boldsymbol{w}_{t+1}\| \cdot \|\boldsymbol{g}_{\boldsymbol{w}_t}^{\boldsymbol{D}_a}\| + \eta\|\Delta\boldsymbol{w}_t\| \cdot \|\boldsymbol{H}_{\boldsymbol{w}_t}^{\boldsymbol{D}_a}\| \cdot \|\boldsymbol{g}_{\boldsymbol{w}_t}^{\boldsymbol{D}_a}\| + \frac{1}{2}\|\Delta\boldsymbol{w}_t + \Delta\boldsymbol{w}_{t+1}\|^2 \cdot \|\boldsymbol{H}_{\boldsymbol{w}_t}^{\boldsymbol{D}_a}\| \tag{21d}$$

$$+ \eta\|\Delta\boldsymbol{w}_t + \Delta\boldsymbol{w}_{t+1}\| \cdot \|\boldsymbol{H}_{\boldsymbol{w}_t}^{\boldsymbol{D}_a}\| \cdot \|\boldsymbol{g}_{\boldsymbol{w}_t}^{\boldsymbol{D}_a}\| + \eta\|\Delta\boldsymbol{w}_t + \Delta\boldsymbol{w}_{t+1}\| \cdot \|\boldsymbol{H}_{\boldsymbol{w}_t}^{\boldsymbol{D}_a}\|^2 \cdot \|\Delta\boldsymbol{w}_t\| + \mathcal{O}(\eta^2)$$

$$\leq \sqrt{n}s_{max}^2 \cdot \|\boldsymbol{g}_{\boldsymbol{w}_t}^{\boldsymbol{D}_a}\| + \frac{1}{2}\sqrt{n}s_{max}^2\eta \cdot \mathrm{Tr}(\boldsymbol{H}_{\boldsymbol{w}_t}^{\boldsymbol{D}_a}) \cdot \|\boldsymbol{g}_{\boldsymbol{w}_t}^{\boldsymbol{D}_a}\| + \frac{1}{8}ns_{max}^4 \cdot \mathrm{Tr}(\boldsymbol{H}_{\boldsymbol{w}_t}^{\boldsymbol{D}_a}) \tag{21e}$$

$$+ \sqrt{n}s_{max}^2\eta \cdot \mathrm{Tr}(\boldsymbol{H}_{\boldsymbol{w}_t}^{\boldsymbol{D}_a}) \cdot \|\boldsymbol{g}_{\boldsymbol{w}_t}^{\boldsymbol{D}_a}\| + \frac{1}{2}ns_{max}^4\eta \cdot \mathrm{Tr}^2(\boldsymbol{H}_{\boldsymbol{w}_t}^{\boldsymbol{D}_a}) + \mathcal{O}(\eta^2)$$

$$= \frac{1}{2}\sqrt{n}s_{max}^2 \cdot \|\boldsymbol{g}_{\boldsymbol{w}_t}^{\boldsymbol{D}_a}\| + \frac{1}{8}ns_{max}^4 \cdot \mathrm{Tr}(\boldsymbol{H}_{\boldsymbol{w}_t}^{\boldsymbol{D}_a}) + \frac{1}{2}\sqrt{n}s_{max}^2 \cdot \|\boldsymbol{g}_{\boldsymbol{w}_t}^{\boldsymbol{D}_a}\| \cdot (1 + 3\eta \cdot \mathrm{Tr}(\boldsymbol{H}_{\boldsymbol{w}_t}^{\boldsymbol{D}_a})) \tag{21f}$$

$$+ \frac{1}{2}ns_{max}^4\eta \cdot \mathrm{Tr}^2(\boldsymbol{H}_{\boldsymbol{w}_t}^{\boldsymbol{D}_a}) + \mathcal{O}(\eta^2) \,.$$

The Eqn. (21d) follows from that the mini-batch $\boldsymbol{B}$ is randomly selected from $\boldsymbol{D}_a$, so the average gradient norms and average Hessian matrices in $\boldsymbol{B}$ are approximate to those in $\boldsymbol{D}_a$. And Eqn. (21e) follows from the upper bound of $\|\Delta\boldsymbol{w}\|$ and the relationship between the trace and eigenvalues of the matrix. $\square$

## B. Experimental Settings

### B.1. Datasets

*Table 4.* Datasets used in our experiments.

| Dataset | Description | Training Set | Test Set | Labels |
|---|---|---|---|---|
| **UTK-Face** | Face image | 18,964 | 4,741 | Age, gender, ethnicity |
| **FER2013** | Facial expression image | 28,708 | 7,178 | Seven facial expressions |
| **CIFAR-10** | RGB image | 50,000 | 10,000 | Ten object classes |
| **Imbalanced-CIFAR-10** | RGB image | 19,375 | 10,000 | Ten object classes |
| **MNIST** | Handwritten-digits image | 60,000 | 10,000 | 0-9 |
| **Imbalanced-MNIST** | Handwritten-digits image | 23,782 | 10,000 | 0-9 |

Table 4 shows all datasets used in this paper. We experiment on the imbalanced UTK-face and FER2013 and the balanced CIFAR-10 and MNIST. Besides we artificially construct two imbalanced datasets Imbalanced-CIFAR-10 and Imbalanced-MNIST by randomly and proportionally discarding images from each class. Details for usage of each dataset are presented in the corresponding subsection in Appendix C.

### B.2. Models and Training Details

The paper adopts the following models to verify the results of the main paper:

- **ResNet18** (He et al., 2016). This model consists of 17 convolution layers, 1 AvgPool layer and 1 fully connected layer and has $\sim 11.7$ million parameters.

- **ResNet50** (He et al., 2016). This model contains 50 convolution layers, 1 AvgPool layer and 1 fully connected layer and has $\sim 25$ million parameters.

- **VGG19** (Simonyan & Zisserman, 2014). This model consists of 19 layers (16 convolution layers, 3 fully connected layers,5 MaxPool layers and 1 SoftMax layer) with $\sim 143$ million parameters.

The datasets and corresponding models used for all experiments in this paper are as follows:

- Experiments on the UTK-Face dataset using the ResNet50 and VGG19 models.

- Experiments on the CIFAR-10 and MNIST datasets using the ResNet18 model.

- Experiments on the FER2013 dataset using the ResNet50 and VGG19 models.

- Experiments on the Imbalanced-CIFAR-10 and Imbalanced-MNIST datasets using the ResNet18 model.

The training process is performed using an NVIDIA 3090Ti device. The hyperparameters for all the models are set with an initial learning rate of 0.001, which is gradually reduced based on the number of epochs during training to optimize the models. The VGG19 model is trained for epochs ranging from 40 to 60, the ResNet18 model also undergoes training for 100 epochs, while the ResNet50 model is trained for approximately 200 epochs.

All experiments are conducted in a Python 3.10 environment using the PyTorch framework. For each experiment, the overall classification accuracy is calculated, along with the classification accuracy on different protected group sets, as well as the gradient norms and the traces of the Hessian matrices. Each experiment is repeated ten times to calculate the average results.

## C. Additional Experimental Results

### C.1. Experiments on (balanced) CIFAR-10 and MNIST using ResNet18

In this section, we conduct experiments on ResNet18 for two balanced datasets CIFAR-10 (Krizhevsky et al., 2010) and MNIST (Deng, 2012), and further support our findings in the main paper by observing the fairness phenomena in balanced datasets. Figure 4 reports our results and shows that as the bit-width gets lower in PTQ and QAT, accuracy gaps between 10 groups on both CIFAR-10 and MNIST is relatively *stable*, while they grow *larger* on UTK-Face in Figure 1. Furthermore, as summarized in Table 1, the fairness metric $\varphi(\boldsymbol{D})$ for the two datasets remains quite small across all bit-widths, indicating the model performs more fairly on these datasets compared to others with higher fairness metric values.

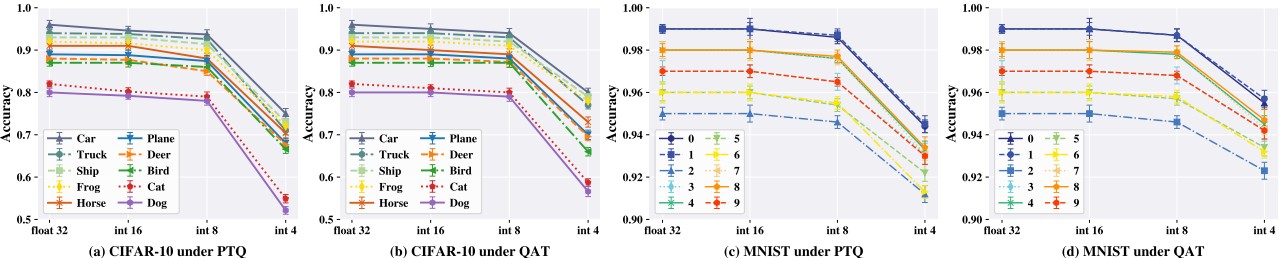

*Figure 4.* Experiments on CIFAR-10 and MNIST datasets on the accuracy of each subgroup of individuals using ResNet18. Both PTQ and QAT are evaluated as bit-widths get lower.

### C.2. Experiments on Imbalanced-CIFAR-10 and Imbalanced-MNIST

To further validate the impact of imbalanced datasets in quantization on the fairness of the model, we manually construct two imbalanced datasets, Imbalanced-CIFAR-10 and Imbalanced-MNIST datasets, based on CIFAR-10 and MNIST datasets, respectively, and experiment on the ResNet50 model.

We construct the Imbalanced-CIFAR-10 dataset by randomly selecting a total of 19375 images from the CIFAR-10 dataset in the ratio of $16 : 16 : 8 : 8 : 4 : 4 : 2 : 2 : 1 : 1$ from the 10 object categories of Plane, Car, Bird, Cat, Deer, Dog, Frog, Horse, Ship and Truck, respectively. Similarly, we construct the Imbalanced-MNIST dataset by randomly selecting a total of

23250 images from the MNIST dataset in the ratio of $16 : 16 : 8 : 8 : 4 : 4 : 2 : 2 : 1 : 1$ from the 10 handwritten digit categories of "0", "1", "2", "3", "4", "5", "6", "7", "8" and "9", respectively.

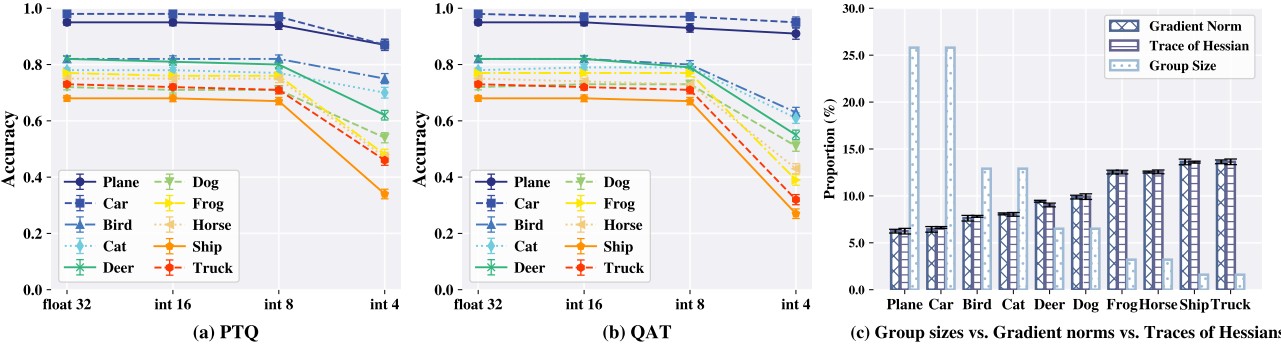

*Figure 5.* (a) and (b) represent the experiments on CIFAR-10 dataset on the accuracy of each subgroup of individuals using ResNet18 for a gender classification task; (c) represents the proportions of gradient norms, traces of Hessian and group sizes for four demographic groups.

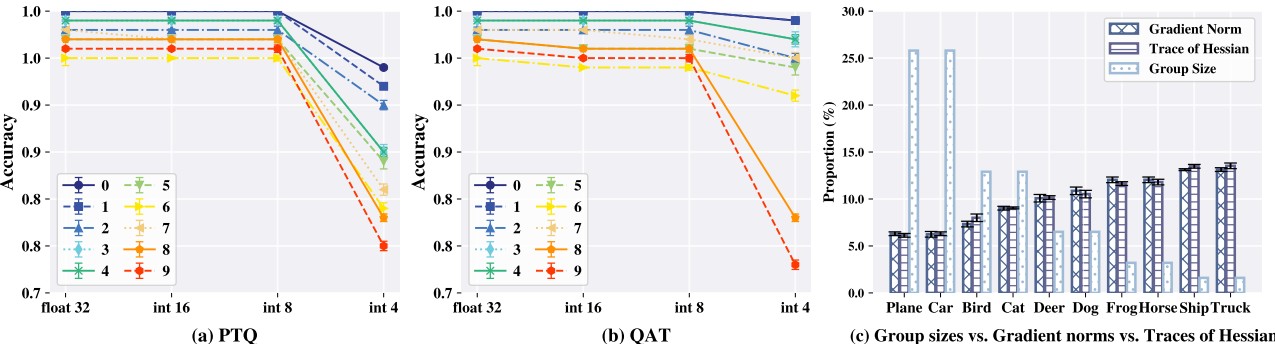

*Figure 6.* (a) and (b) represent the experiments on MNIST dataset on the accuracy of each subgroup of individuals using ResNet18 for a gender classification task; (c) represents the proportions of gradient norms, traces of Hessian and group sizes for four demographic groups.

The experiments in Figure 5 and Figure 6 show the relationships between the gradient norm, the trace of the Hessian matrix, and the model accuracy at three different bit-widths in PTQ and QAT, with Figure 5 corresponding to Imbalanced-CIFAR-10 and Figure 6 to Imbalanced-MNIST. The experimental results also show very similar trends to those reported in the main body of the paper. Taking Figure 5 as an example, groups Frog, Horse, Ship and Truck with larger gradient norms and traces of Hessians compared to the other groups, the accuracy drops off faster. And for a specific bit-width, for example int 4, the accuracy is lower on these groups. Consequently, as the quantization bit-width $b$ decreases, the disparity in $\mathcal{G}(a)$ values across groups becomes more pronounced, leading to an increase in $\varphi(\boldsymbol{D})$ and further exacerbating unfairness.

## C.3. Experiments on different models using different imbalanced datasets

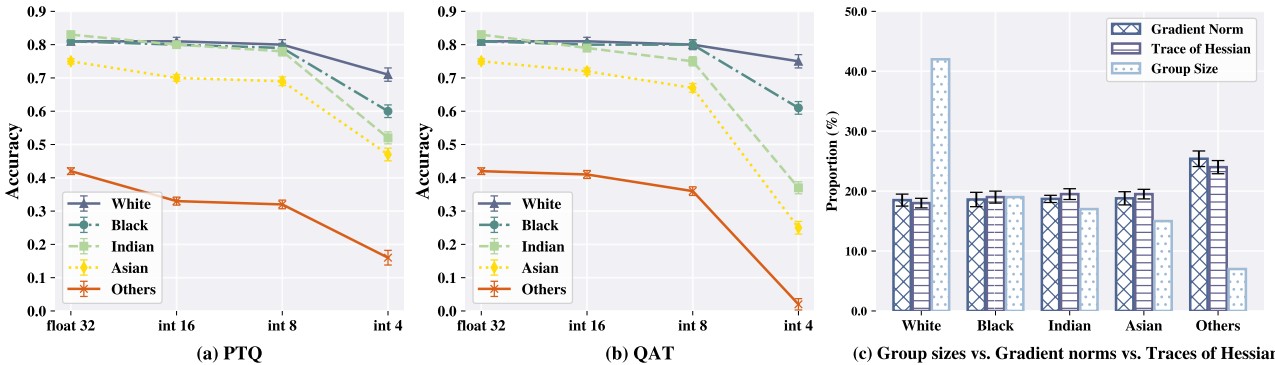

*Figure 7.* (a) and (b) represent the experiments on UTK-Face dataset on the accuracy of each subgroup of individuals using VGG19 for a gender classification task; (c) represents the proportions of gradient norms, traces of Hessian and group sizes for four demographic groups.

To further validate the theoretical and experimental analyses in the main paper, we conduct experiments on different datasets and models to evaluate the impact of quantization on model fairness. In addition to the experiments on the UTK-Face dataset on the ResNet50 model in the main paper, we additionally conduct experiments on the UTK-Face dataset using VGG19 model, and the FER2013 dataset using ResNet50 and VGG19 models.

For the UTK-Face dataset, we set up an ethnicity classification task on the VGG19 model with a protected group that coincides with the target label, i.e., White, Black, Indian, Asian and Others.

For the FER2013(Goodfellow et al., 2013) dataset, we performed facial expressions classification tasks on the ResNet50 and VGG19 models with a protected group set that coincides with the target label set, i.e., Happy, Sad, Neutral, Fear, Angry, Surprise and Disgust.

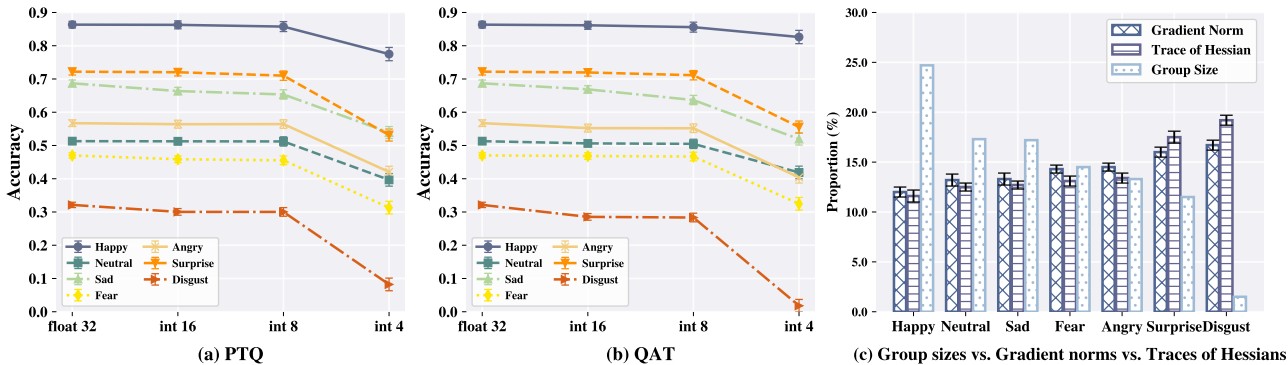

*Figure 8.* (a) and (b) represent the experiments on FER2013 on the accuracy of each subgroup of individuals using ResNet50 for a gender classification task; (c) represents the proportions of gradient norms, traces of Hessian and group sizes for four demographic groups.

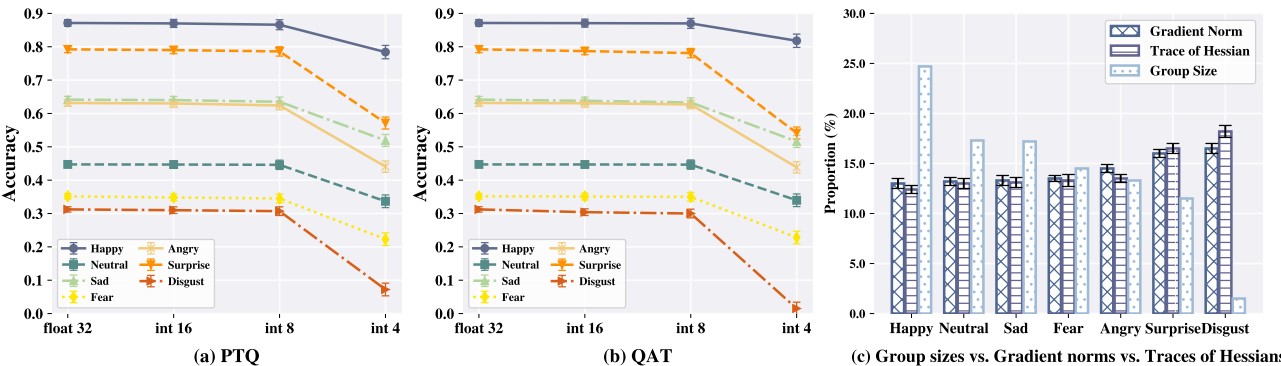

*Figure 9.* (a) and (b) represent the experiments on FER-2013 dataset on the accuracy of each subgroup of individuals using VGG19 for a gender classification task; (c) represents the proportions of gradient norms, traces of Hessian and group sizes for four demographic groups.

The experiments in Figures 7-9 show the relationships between the gradient norm, the trace of the Hessian matrix, and the model accuracy at three different bit-widths in PTQ and QAT. The experimental results show very similar trends to those reported in the main body of the paper: (1) Consider a specific bit-width, for example, int 8 in Figure 7. While the gradient norms and the traces of the Hessian matrices are larger over different groups, e.g., comparing group White and Others, there is an opposite numerical relationship in accuracy; (2) Consider a specific demographic group, for example, group Others in Figure 7. While its gradient norm accounts for the largest percentage, its accuracy decreases the most fast. Similar phenomena happen in traces of the Hessian matrices. As a consequence, the reduction in quantization bit-width $b$ causes the gap in $\mathcal{G}(a)$ values among different groups to widen further, resulting in a higher $\varphi(\boldsymbol{D})$ and thereby intensifying unfairness. These results align with those presented in Table 1.

In a similar manner, to clearly compare the unfairness in PTQ and QAT, Table 1 demonstrates how the fairness metric $\varphi(\boldsymbol{D})$ evolves as the bit-width decreases. As evident, $\varphi(\boldsymbol{D})$ in QAT increases substantially more than in PTQ.

**C.4. Supplementary experiments on mitigation schemes**

**Evaluating our mitigation scheme on different (imbalanced) datasets using different models.**   To further illustrate the effectiveness of GT and RE in mitigating unfairness, We also conduct experiments using ResNet50 and VGG19 on the (imbalanced) FER2013 for both PTQ and QAT as follows:

- Experiments for an ethnicity classification task with ethnicities as the protected attributes on the UTK-Face dataset and VGG19 model. The specific augmentation methods for the five ethnicity groups in UTK-Face are consistent with those described in the main paper.

- Experiments on the ResNet50 and VGG19 models using the FER2013 dataset. The FER2013 dataset consists of 28,708 training images, distributed as follows: 7,091 images for Happy, 4,966 for Neutral, 4,937 for Sad, 4,162 for Fear, 3,818 for Angry, 3,301 for Surprise and 431 for Disgust. To address this imbalance, we augment the training images so that each age group contains 7,091 images.

As shown in Table 5, the values of $\varphi(\boldsymbol{D})$ are considerably reduced, highlighting the effectiveness of data augmentation in mitigating unfairness.

*Table 5.* Fairness metric $\varphi(\boldsymbol{D})$ for VGG19 with data augmentation methods (GT and RE) applied.

| Augmentation Method | Quantization Method | $\varphi(\boldsymbol{D})$ (%) | | | | | |
| --- | --- | --- | --- | --- | --- | --- | --- |
| | | UTK-Face | | | FER2013 | | |
| | | int 16 | int 8 | int 4 | int 16 | int 8 | int 4 |
| Non-Mitigation | PTQ | 11.2 | 12.4 | 54.6 | 0.6 | 1.5 | 50.7 |
| | QAT | 11.7 | 18.9 | 84.1 | 1.4 | 2.9 | 65.7 |
| GT | PTQ | 0.4 | 1.0 | 1.9 | 1.4 | 1.7 | 2.5 |
| | QAT | 1.2 | 1.5 | 2.4 | 1.4 | 1.8 | 3.0 |
| RE | PTQ | 0.5 | 1.2 | 1.5 | 1.7 | 2.0 | 3.7 |
| | QAT | 1.0 | 1.3 | 2.0 | 1.8 | 2.2 | 4.1 |

