# OpenReview forum: "Understanding the Unfairness in Network Quantization"
_ICML.cc/2025/Conference — ICML 2025 poster_

### Official Review · Reviewer_QYq3 · 2025-03-09

**Overall Recommendation:** 2

**Summary:**

This work unveils the potential risk of exacerbating the unfairness in model accuracy among various groups. By theoretical analysis and empirical experiments with both Post-Training Quantization (PTQ) and Quantization-Aware Training (QAT), this work identifies several observations, including group White has less performance drop and PTQ behaves better in preserving fairness. The author also verifies several unfairness mitigation schemes, including geometric transformations and random erasing, and demonstrates that these data augmentation techniques could help mitigate the unfairness caused by quantization.

**Claims And Evidence:**

Most claims made in the submission are well supported by clear and convincing evidence.

**Essential References Not Discussed:**

As many representative works in quantization and fairness are cited, I think the related works section in this paper are essential to understanding the key contributions of the paper.

**Experimental Designs Or Analyses:**

I have checked the soundness/validity of experimental designs. Some issues include:

- The work broadly uses QAT and PTQ to represent the quantization methods used in the analysis. However, these are two major categories instead of the concrete techniques. For QAT, I know from section 3.1 that signed symmetric uniform weight-only quantization, but it's not sure what concrete settings are the author following. I recommend adding the reference to specific methods, such as PACT/LSQ for QAT, or Q-drop/BRECQ for PTQ.
- In addition, this work only verifies the fairness problems under weight-only quantization settings. The generalization of the conclusion on weight-activation quantization remains unknown.

[1] PACT: Parameterized Clipping Activation for Quantized Neural Networks, arxiv 2018

[2] Learned Step Size Quantization, ICLR 2020

[3] QDrop: Randomly Dropping Quantization for Extremely Low-bit Post-Training Quantization, ICLR 2022

[4] BRECQ: Pushing the Limit of Post-Training Quantization by Block Reconstruction, ICLR 2021

**Methods And Evaluation Criteria:**

The proposed methods and evaluation criteria make sense for the problem or application.

**Other Comments Or Suggestions:**

- The title on each page should be revised to be the title of this paper instead of the placeholder "Submission and Formatting Instructions for ICML 2025"
- Some typos could be fixed (Table 1 under QAT int8 ResNet18 CIFAR-10  "01.3" $\rightarrow$ "1.3", line 317 "is" $\rightarrow$ "are", line 423 "adopted on for" $\rightarrow$ "adopted for", )

**Other Strengths And Weaknesses:**

N/A

**Questions For Authors:**

- If the data imbalance is to blame for the unfairness, would data-free quantization methods with balanced synthetic datasets solve the problems?
- From the results shown in Table 1, fairness metrics variation across different models is more significant than in different quantization settings (especially on VGG19). Any ideas or discussions on this phenomenon?

**Relation To Broader Scientific Literature:**

This is a brand new direction and research question. The key contributions are novel, while some fairness metrics and mitigation schemes follow previous work and are cited.

**Theoretical Claims:**

I have checked the correctness of the proofs for theoretical claims in the main text and did not find any issues.

---

> ### Author Rebuttal · Authors · 2025-04-01
>
> Thank you for kindly evaluating that "the key contributions are novel." We also sincerely appreciate your constructive suggestions, and believe that the additional experiments and explanations can address your concerns. The new experimental results are available at https://anonymous.4open.science/api/repo/Rebuttal-30C1/file/Reviewer%20QYq3.pdf?v=f7137c6b.
> - Typos: Thank you for your thorough review and for pointing out these formatting and typographical issues. We have made another careful polish to the paper.
> - Issue 1: To enhance clarity, we specify that our work adopts the quantization method described in [1], which serves as a standard approach in both PyTorch and TensorFlow. This method is widely used in practice and aligns with the default settings in major deep learning frameworks. As suggested, we have also added references to specific quantization methods, including PACT/LSQ for QAT and Q-drop/BRECQ for PTQ, in the Related Work section to improve clarity. Moreover, our theoretical analysis follows general quantization principles and is applicable to different quantization methods (see Table 1 in the link). Specifically, in the final step of the derivation in Theorems 4.1 and 5.1, we only need to replace $\Vert \Delta w \Vert$ with the corresponding quantization error of the specific method—for instance, when applying BRECQ[2], one would replace $\Vert \Delta w \Vert$ with the Frobenius norm of the difference between the full-precision weights and the quantized weights.
> - Issue 2: We would like to clarify that our theoretical analysis for weight-only quantization (WOQ) can be directly generalized to weight-activation quantization (WAQ). This generalization is supported by the findings in [3], specifically Theorem 1, which shows that the impact of activation quantization on the target loss can be transformed into a similar effect as weight quantization. Based on this, we have derived the theoretical bounds for WAQ, which reveal that the upper bound of the excessive loss $G(a)$ introduces two additional terms compared to WOQ: $\sqrt{n}\frac{\tilde{w}^*_{max}s_{max}}{2+s_{max}}\cdot\Vert g_{w^*}\Vert+\frac{1}{2}n\left(\frac{\tilde{w}^*_{max}s_{max}}{2+s_{max}}\right)^2\cdot \text{Tr}(H_{w^*})$. The coefficient $\frac{\tilde{w}^*_{max}s_{max}}{2+s_{max}}$ has a stronger impact on fairness compared to the WOQ case. To further validate this theoretical analysis, we have conducted experiments with WAQ and the results are consistent with our theoretical findings (see Table 2 in the link). We have included these theoretical analyses, proofs, and the corresponding experimental results and discussions in the revised paper.
> - Q1: Data-free quantization methods with balanced synthetic datasets may help mitigate unfairness caused by data imbalance. As a preliminary exploration, we have conducted experiments with GDFQ[4]. Specifically, for a task involving $n$ classes, the generator uniformly samples labels $y∈${$0, 1, \dots, n−1$} during training to ensure a balanced class distribution in the synthetic data. The results show that this approach can mitigate unfairness to some extent (see Table 3 in the link). However, it is important to note that data-free quantization is primarily a technique designed to enable quantization in the absence of training data, rather than a direct solution to address data imbalance. In contrast, data augmentation explicitly targets class imbalance.
> - Q2: We acknowledge that the variation in fairness metrics across different models suggests that, in addition to the two key factors identified in our theoretical analysis, fairness in quantized models may also be influenced by other potential factors, including model architecture. To further study this, we conducted additional ablation experiments to explore more factors affecting fairness in quantized models. Our results indicate that model architecture, optimization algorithm, and hardware selection all play a role, with the VGG architecture, Mini-batch SGD, and Ada L4 GPU particularly exacerbating unfairness (see Table 4-6 in the link). In particular, under int 4, VGG exhibits more severe unfairness than ResNet due to its architectural characteristics. Lacking residual connections, VGG is more prone to gradient vanishing and exploding under quantization, amplifying subgroup gaps. What's more, VGG’s uniform layer structure accumulates quantization errors more significantly, whereas ResNet’s residual connections help mitigate these effects by preserving information flow.
>
> Your insights are valuable to us, and we sincerely appreciate your reconsideration of our paper.
>
> [1] Quantization and Training of Neural Networks for Efficient Integer-Arithmetic-Only Inference, CVPR 2018.
>
> [2] BRECQ: Pushing the Limit of Post-Training Quantization by Block Reconstruction, ICLR 2021.
>
> [3] QDrop: Randomly Dropping Quantization for Extremely Low-bit Post-Training Quantization, ICLR 2022.
>
> [4] Generative Low-Bitwidth Data Free Quantization, ECCV 2020.

---

### Official Review · Reviewer_4FiB · 2025-03-11

**Overall Recommendation:** 3

**Summary:**

They used data enhancement to mitigate the unfairness of quantification of unbalanced data set models

**Claims And Evidence:**

convincing

**Essential References Not Discussed:**

N/A

**Experimental Designs Or Analyses:**

soundness

**Methods And Evaluation Criteria:**

make sense

**Other Comments Or Suggestions:**

My concern is that
1. the takeaway2is obvious such as "Class imbalance is to blame for unfairness" in line 275 and might not contributes significantly.
2. Takeaway 3 "Although quantization-aware training always provides a better overall performance guarantee, deterioration in fairness induced by imbalanced datasets towards protected attributes is much more severe" is  interesting enough but the problems that were found did not seem to be well addressed. Case in tabel 3 when $n$=20 seems to be similar to baseline in Tabel 1.

**Other Strengths And Weaknesses:**

The content of this article is very interesting, however I would suggest that the title be more strictly limited, perhaps in the field of face recognition, racial differences, etc. In addition, the means to reflect the article are solved through data enhancement, that is, the title or abstract needs to reflect the limitations of the datasets used.

**Questions For Authors:**

3. Tabel 2-3 lack the baseline model (Tabel 1) for comparison, making it not straightforward
4. Tabel 3 present a good ablation of problems that were found did not seem to be well addressed, which is not vary stable with different $n$, which should be further discuss.

**Relation To Broader Scientific Literature:**

unbalance data, model compression

**Theoretical Claims:**

correctness

---

> ### Author Rebuttal · Authors · 2025-04-01
>
> Great thanks for kindly evaluating that "the content of this article is very interesting." We also sincerely appreciate your valuable feedback, and believe that the additional experiments, and explanations can address your concerns. The additional experimental results are available at https://anonymous.4open.science/api/repo/Rebuttal-30C1/file/Reviewer%204FiB.pdf?v=27c3e413.
> - Q1: We would like to clarify that our work goes beyond merely stating that "class imbalance causes unfairness" by offering detailed theoretical insights and empirical evidence to deepen the understanding of how class imbalance interacts with quantized model fairness, ultimately guiding the development of effective mitigation strategies. To enhance the clarity and impact of our conclusions, we have revised the manuscript to highlight the novel aspects of our analysis, such as the specific mechanisms by which quantization exacerbates unfairness in imbalanced datasets and the effectiveness of proposed mitigation strategies. To further address your concerns, we have included a new ablation study section after Section 5 to explore potential factors influencing fairness in quantized models. The results indicate that model architecture, optimization algorithm, and hardware selection all have potential impacts on fairness (see Tables 1-3 in the link). Specifically, the VGG architecture, Mini-batch SGD, and Ada L4 GPU tend to exacerbate unfairness in quantized models.
> - Q2: We believe there might be a misunderstanding concerning the purpose of the ablation study in Table 3. The primary aim of this ablation is to investigate how the choice of 𝑛 influences the mitigation of unfairness. In our experiments, the strategies with $n=20$ and $n=3$ proved ineffective, resulting in outcomes closely similar to the baseline reported in Table 1. In contrast, the strategy with $n=10$ closely approach the optimal performance of the proposed method (refer to the results labeled 'RE' in Table 2), effectively mitigating unfairness while narrowing the fairness gap between PTQ and QAT. This sensitivity to the choice of $n$ can be attributed to the impact of mask size on the quality and diversity of augmented data. Larger mask sizes, such as $n=20$, may excessively obscure critical image features, impairing the model's ability to learn. Conversely, smaller mask sizes, like $n=3$, might not provide sufficient diversity to capture the inherent variability of the data. Therefore, an intermediate value like $n=10$ strikes an optimal balance, enhancing both the quality and diversity of augmented data, which leads to more robust and effective fairness improvements. To further substantiate our findings, we have conducted a finer-grained sensitivity analysis on mask size $n$ over the range {$3, 5, 8, 10, 12, 15, 20, 30, 40$} (see Table 4 in the link). We have revised the manuscript to provide a detailed discussion of these results, aiming to clarify any potential misunderstandings.
> - Q3: Thank you for your insightful feedback regarding Tables 2 and 3. To address this, we have revised Tables 2 and 3 in our manuscript to include the baseline model results, facilitating clearer and more direct comparisons.
> - Q4: Great thanks for your positive feedback on the ablation study presented in Table 3. We would like to clarify that the ablation study in Table 3 is not designed to identify the optimal $n$, but rather to validate the superiority of a dynamic, randomized approach over static configurations. Specifically, this ablation demonstrates that the random selection strategy for the mask size $n$—defined within the range {$3, 4, \dots, 20$}, as derived from the optimal configuration in [1]—outperforms fixed choices for $n$.​ Since the selection strategy for appropriate mask size $n$ directly impacts the quality and diversity of augmented data, this findings emphasize that, beyond the amount of augmented data, the quality of augmentation also plays a crucial role in mitigating unfairness. Our results indicate that the random selection strategy not only increases the volume of augmented data but also enhances its diversity, leading to more robust and effective fairness improvements. In contrast, fixed choices for $n$ tend to fail in capturing the inherent variability in the data, resulting in less stable and suboptimal fairness improvements. To further support our conclusions, we have conducted a finer-grained analysis on fixed choices of mask size $𝑛$ over the range {$3, 5, 8, 10, 12, 15, 20, 30, 40$} and compared it to the random selection strategy from the range {$3, 4, \dots, 20$} (see Table 4 in the link). The results show that the random selection strategy is indeed the most effective in mitigating unfairness. We have revised the manuscript to clarify this point more explicitly and have included the additional experiments in the appendix.
>
> We truly value your feedback and are deeply grateful for your continued support.
>
> [1] Random Erasing Data Augmentation, AAAI 2020.

---

### Official Review · Reviewer_c7bP · 2025-03-11

**Overall Recommendation:** 3

**Summary:**

Network quantization, a widely studied model compression method, effectively converts floating-point models to fixed-point models with negligible accuracy loss. Despite its success in reducing model size, it can exacerbate fairness issues across different dataset groups. This paper examines Post-Training Quantization (PTQ) and Quantization-Aware Training (QAT), identifying two primary factors causing these fairness issues through theoretical analysis and empirical verification. The study reveals that while QAT maintains higher accuracy at lower bit-widths, it performs worse than PTQ in terms of fairness. Additionally, simple data augmentation methods can mitigate these fairness issues, especially in cases of class imbalance. Experiments on imbalanced datasets (UTK-Face, FER2013) and balanced datasets (CIFAR-10, MNIST) using ResNet and VGG models validate these findings.

**Claims And Evidence:**

The paper supports its claims through both experimental and theoretical evidence. Both the proofs and experiments effectively fulfill the role of supporting the claims made in the paper. However, some key experiments are included in the appendix.

**Essential References Not Discussed:**

no

**Experimental Designs Or Analyses:**

I think the experiments in appendix should be in main body of paper.

**Methods And Evaluation Criteria:**

The proposed methods and evaluation criteria make sense for the problem

**Other Comments Or Suggestions:**

The content from the quantization to Theorems 4.1 and 5.1 in this paper is excellent, elaborating on the mathematical principles behind the unfairness caused by quantization. However, I find the subsequent analysis of the relationship between the numerical characteristics of datasets and quantization inappropriate. Therefore, I suggest that the author analyze the numerical characteristics of datasets in relation to quantization case by case on different datasets. In the experiments, move the experiments from the appendix to the main body, focusing mainly on the validity of 4.1 and 5.1.

**Other Strengths And Weaknesses:**

The paper offers innovative perspectives and analysis on quantization, particularly addressing the impact of different quantization methods on various classes, which is a novel topic. The detailed analysis and experiments provided for this issue are very convincing. However, the paper has several shortcomings:

1. It lacks further analysis of different methods for Post-Training Quantization (PTQ) and Quantization-Aware Training (QAT).
2. I believe that the experiments in the appendix should be included in the main body of the paper, while some of the current main-body experiments should be moved to the appendix.
3. The conclusion of the paper should focus more on providing theorems rather than analyzing data distribution, as datasets can vary significantly in their distributions. Judgments based on whether a single dataset is balanced or not are not sufficient.

**Questions For Authors:**

1.Even for the same Post-Training Quantization (PTQ), there are currently many quantization methods. It is unclear whether different quantization methods would affect the results of this paper. If possible, please provide proofs and experiments to demonstrate this.
2. For model quantization, besides the quantization of parameters, there is also the quantization of activations. It would be helpful to provide an analysis of how the quantization of activations impacts the results.

**Relation To Broader Scientific Literature:**

This is a new topic

**Theoretical Claims:**

I checked all proof. The proof is simple and clear.

---

> ### Author Rebuttal · Authors · 2025-04-01
>
> Thank you sincerely for commenting that "the detailed analysis and experiments provided for this issue are very convincing." We also truly appreciate your constructive suggestions. We have conducted additional experiments and provided further explanations to address your concerns. The additional experimental results are available at https://anonymous.4open.science/api/repo/Rebuttal-30C1/file/Reviewer%20c7bP.pdf?v=e20ea28c.
> - W1&Q1: Our theoretical analysis follows general PTQ and QAT principles and is designed to be applicable to various quantization methods. Specifically, in the final step of the derivation in Theorems 4.1 and 5.1, we only need to replace $\Vert \Delta w \Vert$ with the corresponding quantization error of the specific method—for instance, when applying BRECQ[1], one would replace $\Vert \Delta w \Vert$ with the Frobenius norm of the difference between the full-precision weights and the quantized weights. These theorems also indicate that the larger the quantization error of a method, the more severe the resulting unfairness. To further address your concerns, we have conducted additional experiments utilizing BRECQ for PTQ and LSQ[2] for QAT in the appendix of the revised manuscript, and the results are consistent with our original findings (see Table 1 in the link).
> - W2&W3&Suggestions: Thanks for your constructive suggestion. Actually, before submitting our manuscript, we faced a dilemma regarding the arrangement of the experiments between the main body and the appendix. In our previous submissions, we primarily focused on investigating the relationship between numerical characteristics of datasets (i.e., gradient norms and Hessian traces) and the fairness of quantized models through case-by-case analyses across different datasets, as you suggested. Moreover, we proposed mitigation strategies by introducing a regularization term in the loss function to penalize differences in gradient norms and Hessian traces across classes.
>
>   However, several previous reviewers questioned the practicality of the extensive discussion on Theorems 4.1 and 5.1. They argued that focusing on the numerical characteristics of datasets lacked intuitive and in-depth insights. Specifically, they pointed out that it was unclear what factors directly influence the gradient norms and Hessian traces, and they suggested that more theoretical and experimental investigation was needed. Additionally, they noted that the mitigation strategies based on gradient norms and Hessian traces, although effective, were impractical for real-world applications due to the high cost.
>
>   To address these concerns, we added further theoretical and experimental analysis of gradient norms and Hessian traces in the main body. We found that class imbalance significantly impacts gradient norms and Hessian traces. Consequently, we replaced the unfairness mitigation strategy with a simpler and more efficient data augmentation approach. These changes from our previous submission led to the current paper.
>
>   We greatly appreciate your suggestion to reconsider the arrangement of the content. In the revised paper, to further strengthen the validity of Theorems 4.1 and 5.1, we have moved the experimental results from Appendix C.4, which demonstrate the validity of these theorems on different datasets, to the main body. Meanwhile, we retained Lemma 4.2, Corollary 4.3, and Lemma 4.4, but relocated their theoretical analysis and experimental verification to the appendix for better clarity and focus.
> - Q2: We would like to clarify that our theoretical analysis for weight-only quantization (WOQ) can be directly extended to weight-activation quantization (WAQ). This extension is supported by the findings in [3], specifically Theorem 1, which shows that the impact of activation quantization on the target loss can be transformed into a similar effect as weight quantization. Based on this, we have derived the theoretical bounds for WAQ, which reveal that the upper bound of the excessive loss $G(a)$ introduces two additional terms compared to WOQ: $\sqrt{n}\frac{\tilde{w}^*_{max}s_{max}}{2+s_{max}}\cdot\Vert g_{w^*}\Vert+\frac{1}{2}n\left(\frac{\tilde{w}^*_{max}s_{max}}{2+s_{max}}\right)^2\cdot \text{Tr}(H_{w^*})$. The coefficient $\frac{\tilde{w}^*_{max}s_{max}}{2+s_{max}}$ has a stronger impact on fairness compared to the WOQ case. To further validate this theoretical analysis, we conduct experiments with WAQ and the results are consistent with our theoretical findings (see Table 2 in the link). We have included these theoretical analyses, proofs, and the corresponding experimental results and discussions in the revised paper.
>
> Your insights are valuable to us, and we appreciate your further surpport a lot.
>
> [1] BRECQ: Pushing the Limit of Post-Training Quantization by Block Reconstruction, ICLR 2021.
>
> [2] Learned Step Size Quantization, ICLR 2020.
>
> [3] QDrop: Randomly Dropping Quantization for Extremely Low-bit Post-Training Quantization, ICLR 2022.

---

### Official Review · Reviewer_Tpv5 · 2025-03-20

**Overall Recommendation:** 2

**Summary:**

The paper investigates the fairness implications of network quantization, focusing on two widely used algorithms: Post-Training Quantization (PTQ) and Quantization-Aware Training (QAT).

The authors identify two key factors that exacerbate unfairness in model accuracy across different groups: the gradient norm of the group loss function and the trace of the group loss function's Hessian matrix. They theoretically analyze and empirically validate these factors, showing that class imbalance leads to distinct values of these factors among different attribute classes, which in turn exacerbates unfairness.

The paper also compares PTQ and QAT, finding that QAT, while generally preserving higher accuracy at lower bit-widths, exacerbates unfairness more severely than PTQ. To mitigate this unfairness, the authors propose and evaluate simple data augmentation techniques, demonstrating their effectiveness in reducing disparate impacts of quantization.

**Claims And Evidence:**

The claims made in the paper are generally well-supported by both theoretical analysis and empirical evidence. The authors provide a detailed theoretical framework to explain how gradient norms and Hessian traces contribute to unfairness in quantized models. They also conduct extensive experiments on multiple datasets (UTK-Face, FER2013, CIFAR-10, and MNIST) and models (ResNet and VGG) to validate their findings. The empirical results align well with the theoretical predictions, showing that groups with smaller datasets experience larger gradient norms and Hessian traces, leading to greater accuracy degradation after quantization.

However, one potential issue is the reliance on synthetic imbalanced datasets (Imbalanced-CIFAR-10 and Imbalanced-MNIST) to validate the impact of class imbalance. While these datasets help illustrate the theoretical points, their artificial nature may limit the generalizability of the findings to real-world scenarios. The authors could strengthen their claims by including more naturally imbalanced datasets (e.g., iNaturalist).

**Essential References Not Discussed:**

None

**Experimental Designs Or Analyses:**

The experimental design is sound and well-executed. The authors conduct experiments on multiple datasets and models, covering both imbalanced and balanced scenarios. They also perform ablation studies to validate the effectiveness of data augmentation techniques in mitigating unfairness. The results are presented clearly, with appropriate visualizations (e.g., accuracy plots, fairness metric tables) to support the findings.

One potential improvement would be to include more detailed ablation studies on the data augmentation techniques. For example, the authors could explore different augmentation strategies or hyperparameters to see how they affect the fairness of the quantized models. Additionally, the authors could provide more insights into why certain augmentation techniques (e.g., geometric transformations vs. random erasing) perform better in specific scenarios.

**Methods And Evaluation Criteria:**

The methods proposed in the paper are appropriate for the problem at hand. The authors use standard quantization techniques (PTQ and QAT) and evaluate their impact on fairness using well-established fairness metrics. The choice of datasets (UTK-Face, FER2013, CIFAR-10, and MNIST) is reasonable, as they cover both imbalanced and balanced scenarios, allowing the authors to demonstrate the impact of class imbalance on fairness.

The evaluation criteria, particularly the fairness metrics are well-defined and appropriate for measuring the disparate impacts of quantization across different groups. The authors also provide a clear explanation of how this metric is derived and why it is suitable for their analysis.

**Other Comments Or Suggestions:**

Please refer to the above comments.

**Other Strengths And Weaknesses:**

**Strengths:**

1. The paper addresses an important and timely issue in machine learning, namely the fairness implications of model compression techniques like quantization.

2. The theoretical analysis is rigorous and provides clear insights into the factors that contribute to unfairness in quantized models.

3. The empirical evaluation is thorough, covering multiple datasets, models, and quantization methods.

4. The proposed data augmentation techniques are simple yet effective, and the authors provide clear evidence of their impact on fairness.

**Weaknesses:**

1. The reliance on synthetic imbalanced datasets (Imbalanced-CIFAR-10 and Imbalanced-MNIST)  may limit the generalizability of the findings to real-world scenarios.

2. While the authors compare PTQ and QAT, they do not explore other quantization techniques (e.g., mixed-precision quantization) that might have different fairness implications.

3. The authors propose data augmentation techniques to mitigate unfairness, but they do not explore other potential mitigation strategies (e.g., reweighting, adversarial training).

**Questions For Authors:**

1. The paper identifies gradient norms and Hessian traces as key factors contributing to unfairness in quantized models. However, are there other potential factors (e.g., model architecture, optimization algorithms) that could also influence fairness in quantized models?

2. The paper focuses on fairness in the context of classification tasks. Have the authors considered whether their findings might extend to other types of tasks, such as generative models?

3. The authors mention that QAT exacerbates unfairness more severely than PTQ due to the interaction between gradient norms and Hessian traces. Could the authors provide more intuition or a simplified explanation for why this interaction leads to greater unfairness in QAT compared to PTQ?

**Relation To Broader Scientific Literature:**

N/A

**Theoretical Claims:**

The theoretical claims in the paper are well-formulated and supported by rigorous proofs. The authors provide detailed derivations for the upper bounds of excessive loss in both PTQ and QAT, and they clearly explain how these bounds relate to the gradient norms and Hessian traces. The proofs are presented in the appendix and appear to be correct, though I did not verify every step in detail.

One minor point is that the authors could provide more intuition or discussion around the theoretical results, particularly for readers who may not be familiar with the mathematical details. For example, explaining why the interaction terms in QAT lead to greater unfairness compared to PTQ could help make the theoretical insights more accessible.

---

> ### Author Rebuttal · Authors · 2025-04-01
>
> We sincerely appreciate your acknowledgment that “the paper addresses an important and timely issue, the theoretical analysis is rigorous, and the empirical evaluation is thorough.” We believe that our experimental results strongly support our theoretical findings. ​In response to your concerns, we have conducted further experiments and confirmed the consistent effectiveness of our method. The detailed results are available at https://anonymous.4open.science/api/repo/Rebuttal-30C1/file/Reviewer%20Tpv5.pdf?v=917ec334.
> - W1: ​Our intention to conduct experiments on Imbalanced-CIFAR-10 and Imbalanced-MNIST is to facilitate **direct** comparisons with CIFAR-10 and MNIST, highlighting the impact of class imbalance on fairness. In addition, we also evaluated our method on UTK-Face and FER2013, two naturally imbalanced datasets. To further address your concern, we have included new experiments on iNaturalist 2017, a real-world highly imbalanced dataset, in the revised version. Our findings remain consistent across all datasets (see Figures 1-2 and Table 1 in the link), reinforcing the real-world generalizability of our conclusions.
> - W2: We claim that our method is applicable to **any quantization precision**, including mixed-precision quantization (MPQ), as demonstrated by our quantization error analysis. While MPQ assigns different bit-widths to different layers or channels, the quantization error at each layer follows the same statistical properties as in uniform-precision quantization. To validate this, we have conducted experiments with MPQ on different models and datasets in the appendix (see Table 2 in the link). We have also added a discussion in Section 3.1 to clarify this point.
> - W3: We confirm that other strategies may also help address unfairness, and we have conducted experiments on reweighting and adversarial training (see Tables 3-4 in the link). Our results show that these methods can indeed help mitigate unfairness. We have expanded the discussion in the appendix. Regarding our choice of data augmentation as a mitigation strategy, our motivation is to provide a **more intuitive validation** of our theoretical conclusion that the exacerbation of unfairness in quantized models is related to class imbalance. To achieve this, we adopted geometric transformation and random erasing at the data processing stage, as these methods have been shown to be both effective in prior research and computationally efficient. In addition, we also explored training-stage strategies by introducing a regularization term in the loss function to penalize differences in gradient norms and Hessian traces across classes. However, this approach is less effective and more costly than data augmentation.
> - Q1: Our study specifically focuses on gradient norms and Hessian traces, both theoretically and experimentally, as they **directly impact** optimization stability and sensitivity to quantization errors. In addition, fairness in quantized models can be influenced by multiple factors. To further investigate this, we have included a new ablation study in the appendix that explores additional factors influencing fairness in quantized models. The results indicate that model architecture, optimization algorithm, and hardware selection all have potential impacts on fairness (see Tables 5-7 in the link). Specifically, the VGG architecture, Mini-batch SGD, and Ada L4 GPU tend to exacerbate unfairness in quantized models.
> - Q2: We believe that our fairness measurement method can be **directly generalized** to generative tasks, as it relies on the loss difference before and after quantization, with the cross-entropy loss in classification replaced by the appropriate loss function for generative models. To validate this, we have conducted experiments on VAE in the revised paper, using Evidence Lower Bound (ELBO) loss as the evaluation metric (see Table 1 in the link). Our results indicate that the fairness trends observed in classification tasks remain consistent in generative models, further supporting the generalizability of our findings.
> - Q3: Thank you for your valuable suggestion. To provide more intuition, QAT exacerbates unfairness more than PTQ due to the dynamic interaction between gradient norms and Hessian traces under quantization constraints. Since QAT applies quantization throughout training, gradient updates must adapt to quantization-induced noise, leading to optimization in a more distorted loss landscape. In regions with high Hessian traces, the steep loss surface amplifies the effect of large gradient norms, causing uneven updates across subgroups. In contrast, PTQ quantizes only after full-precision training, avoiding these interaction effects and resulting in relatively lower unfairness. We have added a more detailed discussion in the revised paper and believe this addition improves the accessibility of our theoretical insights.
>
> We greatly appreciate your thorough review and thank you for reconsidering our paper.

---

### Decision · Program_Chairs · 2025-05-01

**Decision:**

Accept (poster)

**Comment:**

This paper studies Post-Training Quantization (PTQ) and Quantization-Aware Training (QAT) to understand how they affect model accuracy across different groups of datasets. It provides both theoretical analysis and empirical verification. The reviewers found the results interesting but provided some comments, especially on the experimental settings and presentation, which need to be addressed.